# Influence of Blade Fracture on the Flow of Rotor-Stator Systems with Centrifugal Superposed Flow

**Gang Zhao** [1] , **Tian Qiu** [2] **and Peng Liu** [2,*]

1 School of Energy and Power Engineering, Beihang University, Xueyuan Road No. 37, Haidian District, Beijing 100191, China; sy1404313@buaa.edu.cn
2 Research Institute of Aero-Engine, Beihang University, Xueyuan Road No. 37, Haidian District, Beijing 100191, China; qiutian@buaa.edu.cn
* Correspondence: liupeng91@buaa.edu.cn

**Abstract:** Rotor-stator cavities are often found in turbomachinery; they supply cold air that is bled from the compressor to the turbine blades. The pressure of the outlet of a rotor-stator cavity is axisymmetric under normal circumstances. However, its pressure would be non-axisymmetric in the event of blade fracture. The impact of blade fracture on a rotor-stator cavity with centrifugal superposed flow is studied in this paper. The Euler number $E$, the rotational Reynolds number $Re_\varphi$, and the low-pressure zone range $\theta$ are investigated and, for the first time, with the non-axisymmetrical boundary conditions employing numerical simulation. The results of the numerical calculations show that after turbine blade fracture, the velocity is more affected in the downstream region at a high radius, especially when the $Re_\varphi$ is large. As for the distribution of the mass flow rate, there may be a critical $\theta_c$ at which the other blades are least affected. The $\theta_c$ would increase as the $Re_\varphi$ or the $E$ increase, and the $\theta_c \cong 0.2$ when $C_w = 10,137$, $Re_\varphi = 5.12 \times 10^5$, and $0.2 \le E \le 0.4$. In addition, the thrust coefficient increases as the $E$ or the $\theta$ increases, and the increase in the thrust coefficient does not exceed 4% when the $E = 0.2$ and the $\theta = 0.1$ in this paper. However, the moment coefficient on the rotating shaft is almost independent of the $E$ and the $\theta$. An increase in the $Re_\varphi$ will reduce the effect of turbine blade fracture on the thrust and moment coefficients, when the $Re_\varphi$ is small.

**Keywords:** rotor-stator system; non-axisymmetric boundary conditions; numerical simulation; turbine blade fracture

## 1. Introduction

By raising turbine-entry temperatures, some axial turbomachines may reach an efficiency of approximately 80% with well-designed impellers. While part of the temperature increase may be attributed to the discovery of new materials, the majority would be attributed to advancements in cooling technologies. For this purpose, a small amount of the air collected from the compressor is utilized to cool the nozzle guide vanes, turbine blades, and disks. This type of flow may be found in almost all turbomachinery cavities positioned between the rotating impeller and the stationary disk, which is called the rotor-stator system.

Concerns about the design of turbines are divided into two categories: blade design and rotor-stator cavity design. The design of a rotor-stator cavity is critical, since it is directly connected to numerous practical concerns, such as swirl ratio, axial thrust, and moment coefficients. To fulfill industrial expectations, the causes of axial thrust and disk frictional loss are explored.

The core swirl ratio $\beta$ (the ratio of the fluid's angular velocity to that of the disk at half the axial gap width) is used to demonstrate the fluid's dominating tangential motion. The pressure distribution along the rotor may be approximated using the core swirl ratio. The axial thrust operating on the disks may be calculated using the pressure distribution. In addition, the moment coefficient is affected by the tangential velocity profile. Thus,

it can be seen that the core swirl ratio is one of the most important parameters in the design of a turbine.

In the case of an enclosed rotor-stator cavity, the core swirl ratio depends exclusively on geometry. Daily and Nece [1] classify the flow regimes into four categories according to core swirl ratio, and turbulent flow regimes (regimes III and IV) are more likely to occur in turbines. Will [2] summarizes some of the usual $\beta$ values reported in the literature and uses $\beta = 1/\left(1 + \sqrt{1 + 5G}\right)$ to forecast the effect of $G$ on $\beta$ for enclosed rotor-stator cavities.

As for the impact of superposed flow on rotor-stator cavity, Owen [3] provides a flow model to determine the core swirl ratio by solving the Ekman equations for flow regime IV. In his configuration, the flow direction is usually radially outward. The Ekman equations are linearized, simplified versions of the equations of motion. Finally, the functional relation for the core swirl ratio in laminar flow is derived by ignoring the outer cylindrical wall in the motion equation. Owen [3] used the integral technique to solve the nonlinear equations of motion for the disk boundary layer in a turbulent flow. The overall decelerating effect of an outer (stationary) shroud on core rotation may be seen in both measured and calculated data. Will [2] introduced a correction for the stator's friction factor in order to reflect such effects in the flow model. Please see the review literature [2–4] for further information.

When evaluating the above works from a safety perspective, these efforts that can minimize the likelihood of failure might be referred to as "active safety designs." Despite significant attempts to protect the blade from high strain, the service experience of aero engines suggests that turbine blade fracture is a common problem [5,6]. As a result, substantial attention is devoted to secondary failure induced by initial turbine blade fracture, with an emphasis on failure development. Then, to avoid and manage the spread of failure, many designs were developed that are examples of "passive safety designs." This article focuses on the influence of turbine blade fracture on the engine, which is a concern of "passive safety designs."

The outlet boundary of the rotor-stator cavity is axisymmetric in the case of common conditions (that is, the backpressure of the cavity is uniform). However, when turbine blades fracture in actual work, the channel inside the broken blades is exposed to the low-pressure mainstream, which results in discrepant pressure at the outlet of the rotor-stator cavity. Because of the discrepant pressure, a large volume of cold air will rush to the broken blades, reducing the cold air acquired by the normal blades, assuming that the total volume of cold air is constant. A normal blade may potentially fail, due to overlimiting the thermal stress, because the amount of cold air obtained by the normal blade is less than the needed quantity. A continuing development is that the distribution of cold air will become increasingly uneven, and eventually cause a cascade of turbine blades to break [7]. In addition, the probability of turbine blade fracture exceeds $10^{-7}$ times per flight hour [8], which is above the probability of hazardous occurrences [9] required by FAA AC 33.75. Therefore, the influence of blade fracture on gas turbines must be demonstrated. In previous publications, the impact of blade fracture on turbines has been investigated [8]. This paper focuses on the influence of the non-axisymmetric boundary caused by turbine blade fracture on a simple rotor-stator cavity.

The axisymmetric boundary and the axisymmetric configuration were used in previous studies of the rotor-stator cavity. Few papers focused on the flow with non-axisymmetric boundary conditions in a rotor-stator system. Below is a synopsis of the relevant research.

Bein et al. [10,11] explored a lubricating oil sealing problem in a narrow rotor-stator cavity. At the outlet of the cavity, there is a high-pressure zone and a low-pressure zone, and a certain pressure lubricating oil is provided at the disk center. The problem has the following characteristics: first, the rotational Reynolds number is very small, $Re_s = \Omega s^2/\nu \ll 1$; second, the Euler number is sufficiently large, $E = (p_2 - p_1)/0.5\rho\Omega^2 b^2 \gg Re_s{}^{-1}$; third, the gap is narrow, $G \ll 1$. Based on these assumptions and characteristics, the velocity and pressure can be obtained by solving the simplified NS equation.

Another type of problem is gas ingress. There has been abundant research on this problem [12,13]. This paper only refers to one significant theoretical model for analyzing

this issue: the orifice model. The orifice model was proposed by Owen [14]. It is based on two assumptions: (1) that the flow is from a large reservoir through a small nozzle; and (2) that there is a discontinuous flow across an imaginary surface of the actuator disk. For an axial clearance seal, the orifice model is built on an imaginary "orifice ring." Through the tiny regions $\delta A_e$ and $\delta A_i$, which add up to the clearance area $A_c$ of the seal, egress and ingress occurs across the separate regions of the orifice ring at the same time. For the inviscid equations, mass and energy are considered to be continuous inside the separate stream tubes for egress and ingress, but there is a pressure discontinuity across the orifice ring. The principal "orifice assumptions" are that $(r_2 - r_1)/r_1 \ll 1$ and $V_{r,1}^2 \ll V_{r,2}^2$ for egress, and vice versa for ingress. When the pressure distribution inside and outside the orifice ring is known, the velocity distribution of ingress and egress can be solved via the orifice model.

For the rotor-stator cavity studied in this paper, the $Re_\varphi \gg 1$ and the inertial force term cannot be ignored, so the model of Bein et al. [10] cannot be used. Additionally, because the radial velocity difference between different radii in the cavity is so small, the orifice model cannot be utilized. This paper investigates the effect of turbine blade fracture on the rotor-stator cavity, employing numerical simulation in terms of swirl ratio, mass flow rate distribution, thrust, and moment coefficients.

## 2. Computational Setup

A typical diagram of a rotor-stator system is shown in paper [15], where the radius of the nozzles is lower than that of the receiver holes and the cold air flow is radially outward. Generally, when a turbine blade fractures, structures such as the mortise are intact. Therefore, this can lead to a low outlet pressure zone in the rotor-stator system. To investigate the effect of turbine blade fracture on a rotor-stator system in isolation, a simple disk cavity model, as shown in the Figure 1, was selected for this paper.

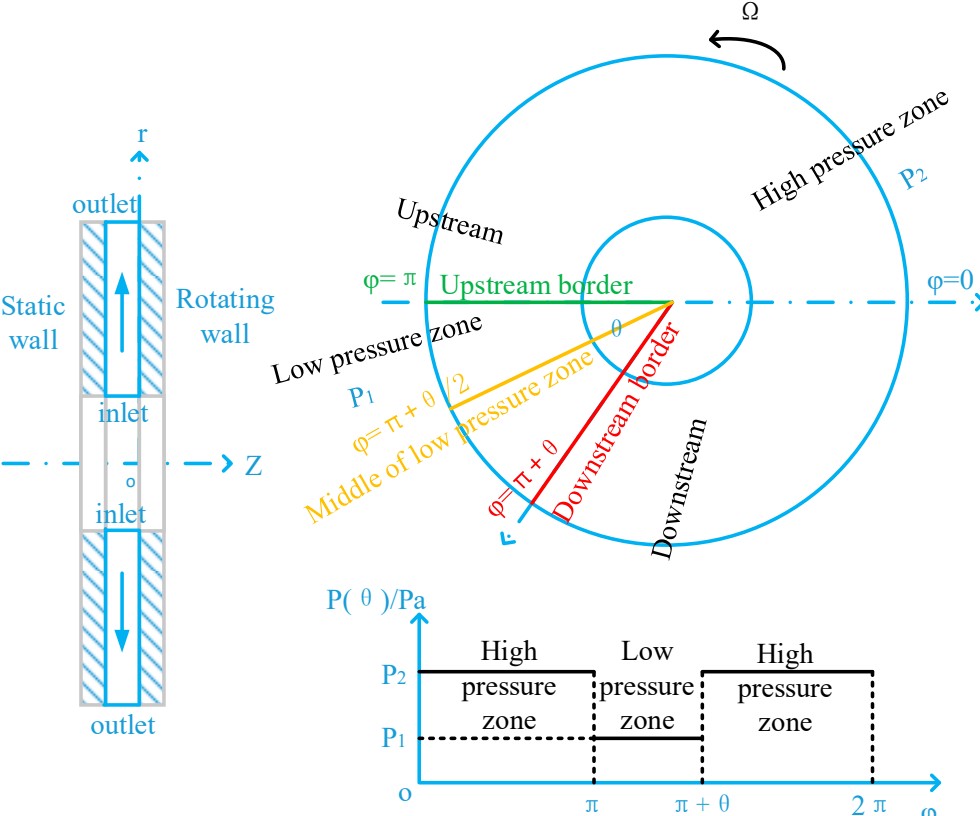

**Figure 1.** Computational model used in this paper.

When turbine blades fracture, the backpressure of the outlet in the rotor-stator cavity is no longer axisymmetric, assuming that the backpressure profile of the outlet is a stepped distribution, as shown in Figure 1, $P(\theta)$. The high-pressure zone has a pressure of $P_2$, and the low-pressure zone has a pressure of $P_1$. The low-pressure zone covers the area where the turbine blades are fractured and the high-pressure zone covers the area where the blades are intact. The upstream area is located in the counter-rotating direction of the low-pressure zone, while the downstream area is located in the cis-rotating direction of the low-pressure zone. The upstream border and the downstream border are the interfaces between the high- and low-pressure zones, where the former is located at $\varphi = \pi$ (shown as the green line in Figure 1) and the latter is located at $\varphi = \pi + \theta$ (shown as the red line in Figure 1). The middle of the low-pressure zone means the position in the middle of the low-pressure zone, located at $\varphi = \pi + \theta/2$ (shown as the yellow line in Figure 1). A sector calculation domain cannot be adopted because the boundary conditions are not axisymmetric. The calculation uses a simple 360°domain without a peripheral wall, as shown in Figure 1. The flow enters radially at a low radius and exits at a high radius, and the coordinate is fixed at the rotor's center. The direction of rotation is pointing to the stator and is perpendicular to the rotor. The gap between the rotor and the stator is 9 mm, and the radius of the rotor and stator is 250 mm.

Poncet et al. [16–18] found that when the flow rate is modest, the flow structure in a rotor-stator cavity with radial outflow is similar to that of a closed rotor-stator cavity, while the flow structure becomes Stewartson type when the flow rate is large. Combined with the research of Owen et al. [3,19], the turbulence parameter $\lambda_T$ affects the flow structure in the cavity without pre-swirl. A Stewartson-type structure is found at a low radius and a Batchelor-type structure is found at a high radius in the case of radial outflow. The Stewartson region at a low radius will expand as the $\lambda_T$ increases, while the Batchelor region will shrink until the Stewartson region is filled throughout the entire cavity. Therefore, the $\lambda_T$ is a vital parameter in the case of an intact turbine (in axisymmetric boundary conditions). We also wanted to know what role it plays in the case of a non-intact turbine (in non-axisymmetric boundary conditions). Therefore, the $\lambda_T$ covers the typical working condition of a rotor-stator cavity [3,20] in this paper. Bein et al. [10,11] employed the Euler number, $E$, to describe the axisymmetry of boundary conditions relative to centrifugal effect. This paper also adopts the $E$, assuming that the pressure at the mortise (outlet of the cavity) equals the mainstream pressure after the turbine blades fracture. The backflow margin of the film hole is roughly 15% to 20% under design conditions [21]; therefore, the pressure ratio between the high-pressure zone and the low-pressure zone is about $1 \leq P_2/P_1 \leq 1.3$ (assuming that the total pressure relative to the turbine rotor is equal to the static pressure). Considering the range of the rotational Reynolds number, the Euler number is $0 \leq E \leq 0.4$ in this paper.

Another important parameter is the range of low-pressure, denoted as $\theta$, which describes the number of broken blades. Modern gas turbines typically have 60–70 turbine blades, so a single turbine blade accounts for approximately 0.1 rad. A single turbine blade fracture results in a low-pressure zone range of 0.1 rad, and a succession of N blade fractures results in a low-pressure zone range of 0.1 * N rad, using $0 \leq \theta \leq 0.4$ in this paper. The cases and parameters involved in this paper are shown in Table 1. When the $E$ or the $\theta$ is zero, the turbine blades are intact and the boundary conditions are axisymmetric (corresponding to cases I, J, K, L in Table 1).

**Table 1.** The range of parameters.

| Case | $G$ | $C_w$ | $\Theta$ | $Re_\varphi$ | $\lambda_T$ | $E$ |
|------|------|--------|-----|---------------|--------|------|
| A | 0.036 | 13,107 | 0.1 | $2.39 \times 10^5$ | 0.65 | 0.2 |
| B | 0.036 | 13,107 | 0.1 | $5.12 \times 10^5$ | 0.35 | 0.2 |
| C | 0.036 | 13,107 | 0.1 | $1.02 \times 10^6$ | 0.20 | 0.2 |
| D | 0.036 | 13,107 | 0.1 | $2.39 \times 10^6$ | 0.10 | 0.2 |
| E | 0.036 | 13,107 | 0.1 | $5.12 \times 10^5$ | 0.35 | 0.4 |
| F | 0.036 | 13,107 | 0.1 | $5.12 \times 10^5$ | 0.35 | 0.04 |
| G | 0.036 | 13,107 | 0.2 | $5.12 \times 10^5$ | 0.35 | 0.2 |
| H | 0.036 | 13,107 | 0.4 | $5.12 \times 10^5$ | 0.35 | 0.2 |
| I | 0.036 | 13,107 | 0 | $2.39 \times 10^5$ | 0.65 | 0 |
| J | 0.036 | 13,107 | 0 | $5.12 \times 10^5$ | 0.35 | 0 |
| K | 0.036 | 13,107 | 0 | $1.02 \times 10^6$ | 0.20 | 0 |
| L | 0.036 | 13,107 | 0 | $2.39 \times 10^6$ | 0.10 | 0 |

The grid used in this paper was meshed by rotating and copying a 2D planar grid to eliminate the effects of a non-axisymmetric grid on calculation (meshed by ICEM19.1). The local grid is shown in Figure 2. Grid setting refers to [22], with 200 radial grids, 140 axial grids, 360 tangential grids, a 0.001 mm thickness of the first layer in the wall boundary layer, and a 1.1 growth rate. The total number of grids surpasses ten million, and $Y^+ < 1$ in almost all the disks in the above cases.

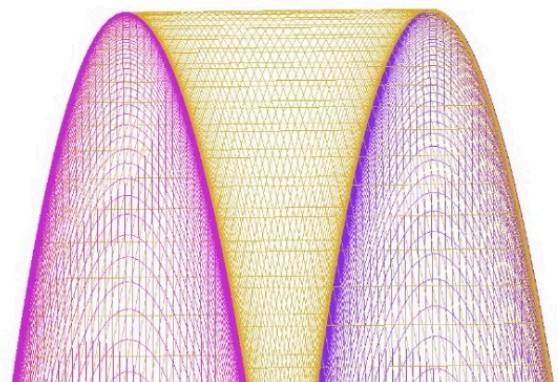

**Figure 2.** Local mesh.

The working medium was air with constant density, viscosity, and heat capacity, calculated by commercial software, CFX19.1. A nonslip adiabatic wall condition was defined for all walls. According to the operating conditions in the pre-swirl system, the wall was set to be stationary or rotating, and the rotational speed was set for rotor. The turbulence model of SST and a high-order discrete scheme were employed in this article, according to Poncet et al. [22] and Da [23]. To ignore the effect of heat transfer on the flow, the air was set as an isothermal fluid. Computational convergence was accepted when the residual of the continuity equation converged to $RMS < 10^{-6}$.

## 3. Results and Discussion

### 3.1. Swirl Ratio

To calculate the axial thrust, it was first necessary to calculate the distribution of the pressure across the disk surface. The distribution of pressure was, in turn, highly dependent on the flow structure in the cavity. Based on previous studies [1], the flow structures were classified by different distributions of the swirl ratio in the axial direction. Therefore, the effect of turbine blade fracture on the swirl ratio was analyzed initially, as set out below.

The distribution of the $\beta$ in the radial direction for different circumferential positions is shown in Figure 3. As can be seen from that figure, $\beta$ at $\phi = 0$ when blade fracture is

exactly the same as when the blades are intact. This means that the position away from the low-pressure zone was not affected by turbine blade fracture in terms of the $\beta$. Furthermore, in the low-radius region ($r^* < r_c^* \cong 0.7$), the $\beta$ values at different circumferential positions were equal, implying that the effects of turbine blade fracture did not propagate to the low-radius position. In the high-radius region ($r^* > r_c^* \cong 0.7$), on the other hand, the $\beta$ was affected to a greater extent. Specifically, in the upstream half of the low-pressure zone ($\phi < \pi + \theta/2$), there was a larger $\beta$ when blades fractured, while in the downstream half of the low-pressure zone ($\phi > \pi + \theta/2$), the $\beta$ was smaller. Clearly, $r_c^*$ decreases as the degree of boundary asymmetry increases (i.e., the $E$ or the $\theta$ increases). It is also worth noting that the $\beta$ values at the upstream and downstream borders of the low-pressure zone ($\phi = \pi$ and $\phi = \pi + \theta$) were almost equal in magnitude and opposite in sign, with the absolute value of the $\beta$ at the downstream border ($\phi = \pi + \theta$) being somewhat larger. The reason for the rapid zeroing of the $\beta$ near the outlet is due to the stepped outlet pressure. Figure 4 shows the velocity vector and contour of the $\beta$ when the $Re_\varphi = 5.12 \times 10^5$, $E = 0.2$, $\theta = 0.1$ at $z/s = 0.5$, from which it can be seen that the $\beta$ is roughly symmetrically distributed. However, at locations very close to the outlet ($r/b = 1$), the small amount of fluid near the upstream and downstream borders is squeezed by the fluid at the lower-radius locations, causing it to be unable to flow out from the outlet of the low-pressure zone and only be able to flow radially at a lower velocity. As a result, the $\beta$ tends to zero at these extreme locations and, again, these locations are more likely to experience backflow.

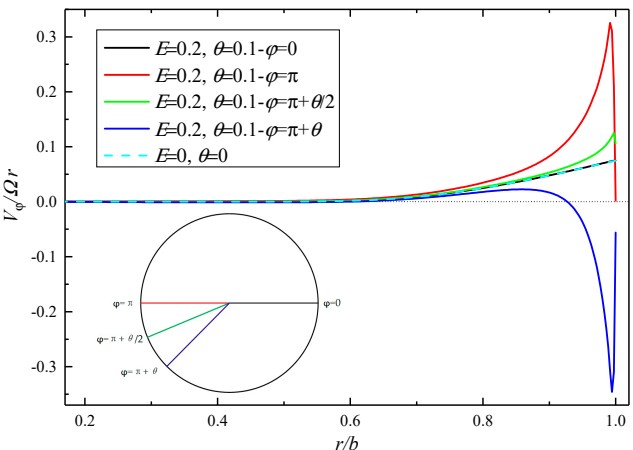

**Figure 3.** The $\beta$ in different circumferential positions when the $Re_\varphi = 5.12 \times 10^5$ at $z/s = 0.5$.

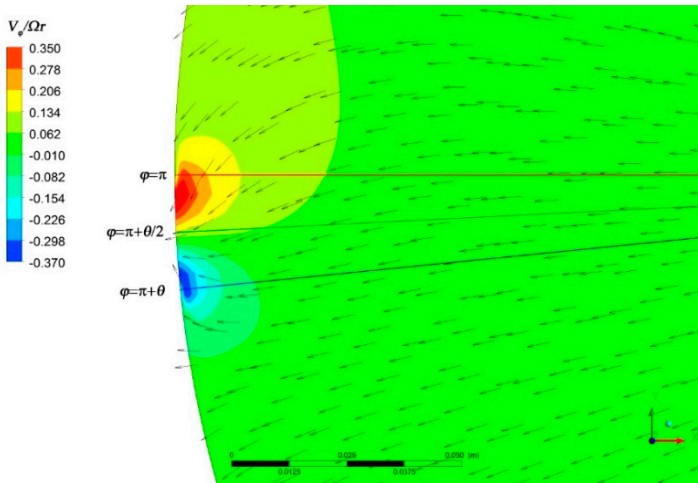

**Figure 4.** Velocity vector and contour of the $\beta$ when the $Re_\varphi = 5.12 \times 10^5$, $E = 0.2$, $\theta = 0.1$ at $z/s = 0.5$.

Figure 5 shows the distribution of the $\beta$ in the circumferential direction for different radii. From preliminary calculations, it can be concluded that the rate of change of the $\beta$ relative to that of an unfractured blade hardly varies with the radius after turbine blade fractures. Furthermore, as the $Re_\varphi$ increases (from case A/I to D/L), the downstream area of the low-pressure zone is progressively more affected than the upstream area.

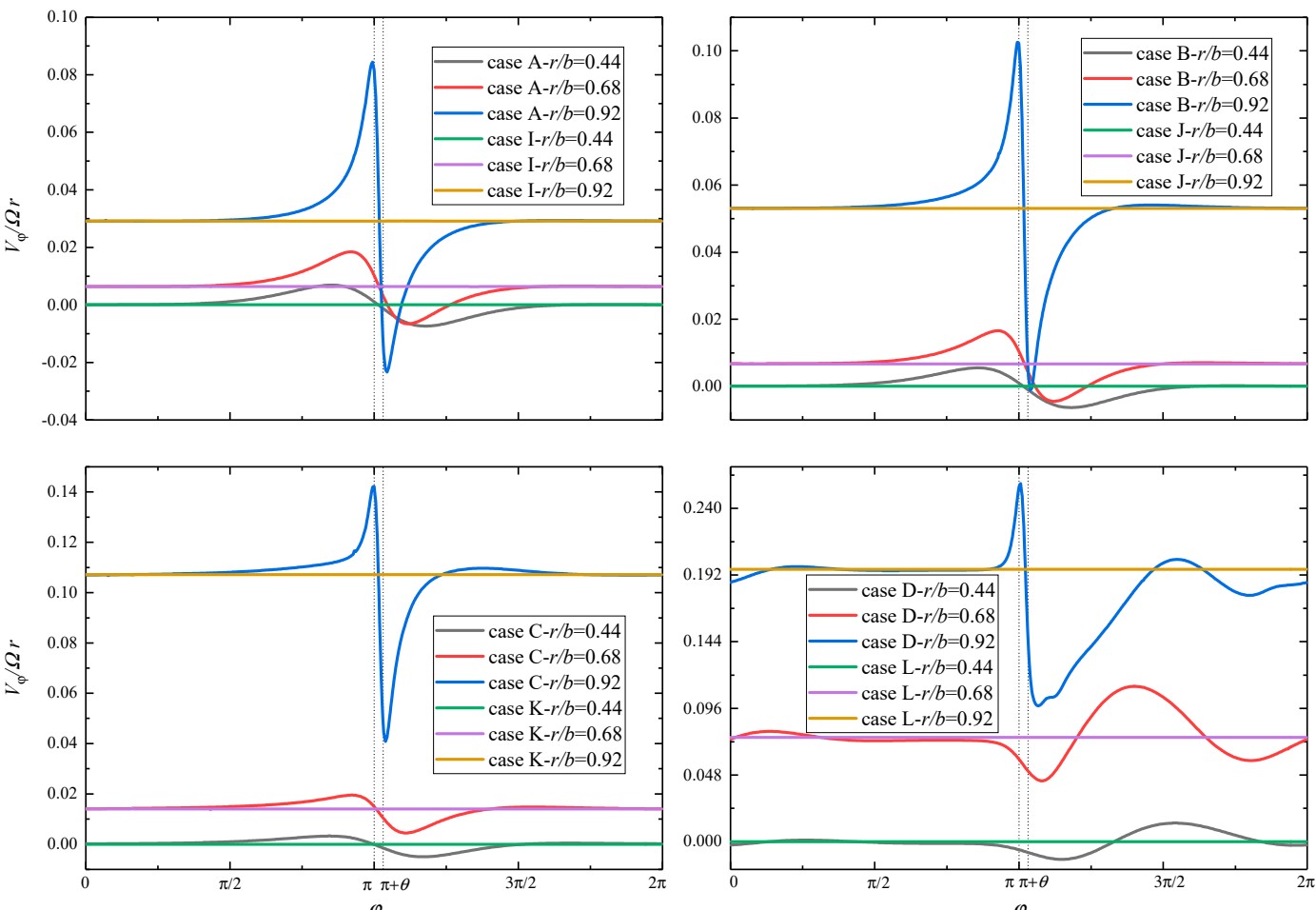

**Figure 5.** $\beta$ curves with different radius at $z/s$ = 0.5.

Another point of interest is that as the $Re_\varphi$ increases, the rate of change of the $\beta$ decreases (see Figures 6 and 7). For example: at $r/b$ = 0.92, the maximum $\beta$ increases by 180%, 102%, 62%, and 50% when $Re_\varphi = 2.39 \times 10^5$, $5.12 \times 10^5$, $1.02 \times 10^6$, and $2.39 \times 10^6$, respectively. This is because the influence of centrifugal forces becomes greater as the $Re_\varphi$ increases and the circumferential imbalance is gradually suppressed by the centrifugal forces, so that the variations in the $\beta$ decrease. However, as the $Re_\varphi$ continues to increase, the centrifugal forces, which are still gradually increasing, slowly plateau. As a consequence, the rate of change in the $\beta$ gradually tends to a constant.

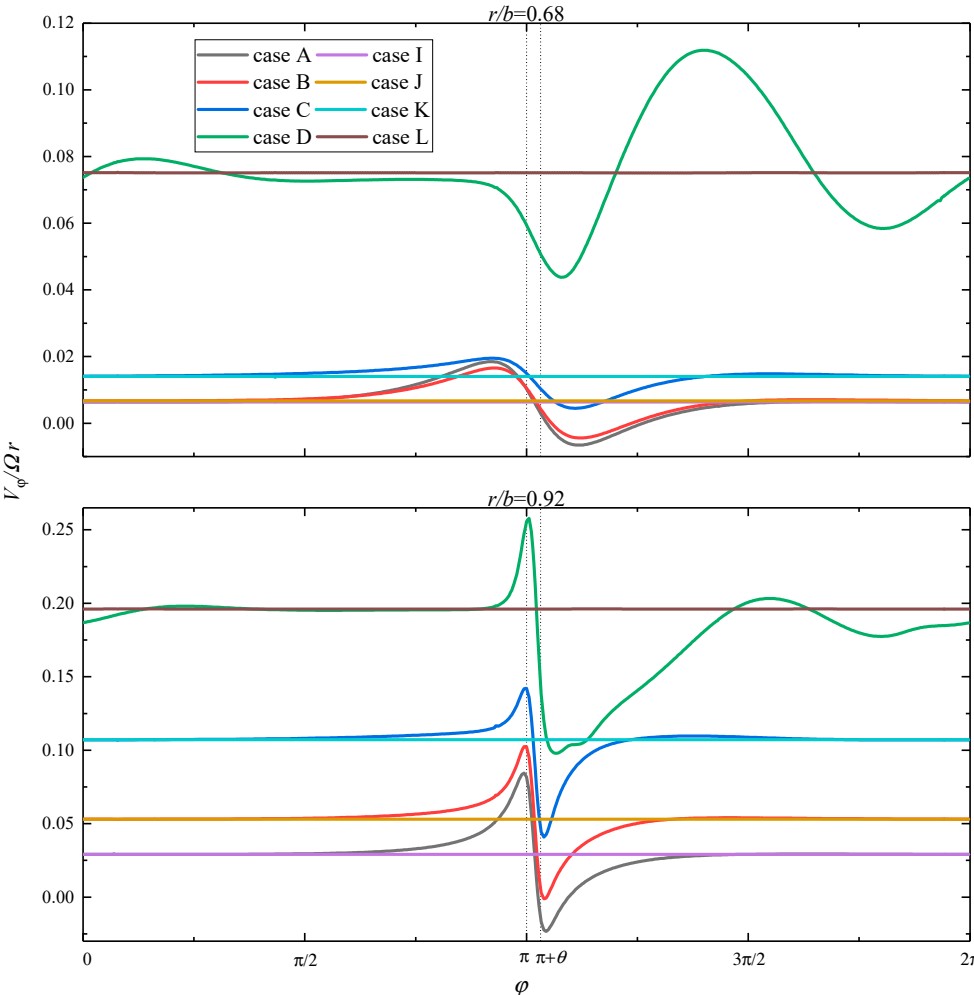

**Figure 6.** $\beta$ curves with different $Re_\varphi$ for $C_w = 10,137$, $E = 0.2$, $\theta = 0.1$ at $z/s = 0.5$.

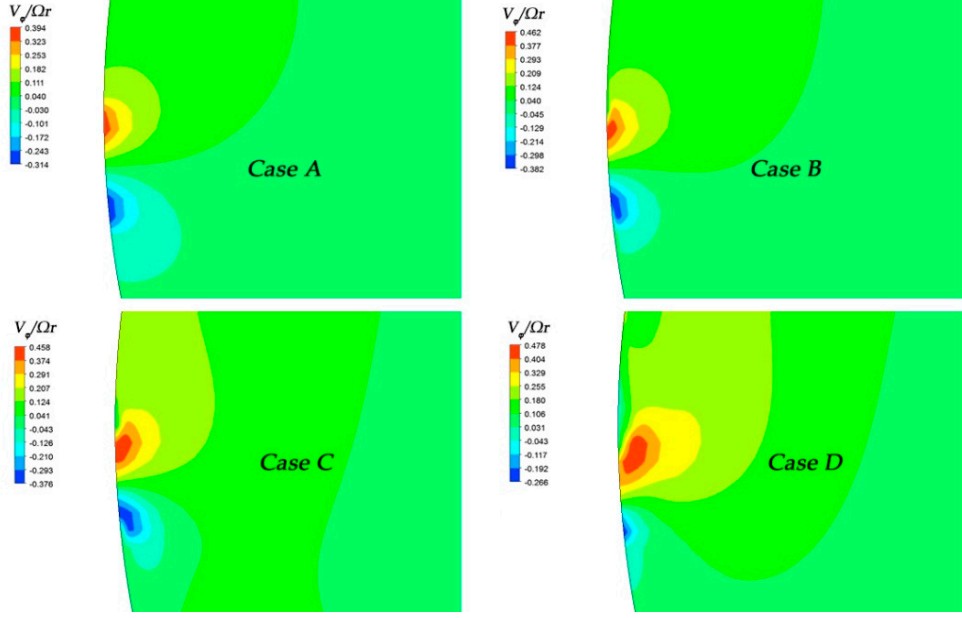

**Figure 7.** Contour of the $\beta$ when $E = 0.2$, $\theta = 0.1$ at $z/s = 0.5$.

The distribution of the $\beta$ in the circumferential direction for different Euler numbers and different $\theta$ numbers is shown in Figures 8 and 9, respectively. The dashed vertical lines in the figures represent the position of the downstream border of the low-pressure zone at different $\theta$ numbers. It can be seen from the two figures that the effect of the $E$ and the $\theta$ on the $\beta$ after turbine blade fracture is similar. As the $E$ or the $\theta$ increases, the fluctuation of the $\beta$ increases, especially at high-radius locations. The standard deviation of the $\beta$ at $r/b = 0.44$, 0.68, and 0.92 is 0.00381, 0.00576, and 0.01612, respectively, when $E = 0.4$ (case E in Figure 8). It is worth noting that the change in the maximum and minimum values of $E \neq 0$ or $\theta \neq 0$, relative to that of $E = 0$ or $\theta = 0$, was not very different, and is slightly larger downstream, which was probably due to the coriolis force of the counter-rotational direction. It is known that when the rotor is stationary, the $\beta$ will be distributed symmetrically by the middle of the low-pressure zone. However, when the rotor has a rotational speed, the fluid near the upstream and downstream borders is subjected to a reversing tangential coriolis force. As a result, the negative $\beta$ at downstream borders is smaller, while the positive $\beta$ at upstream borders is somewhat reduced. Another interesting point in Figure 9 is that at high radii ($r/b = 0.92$), the peaks of both negative and positive $\beta$ appearing around the upstream and downstream borders are shifted towards the middle of the low-pressure zone as the $\theta$ increases, compared to a symmetrical case.

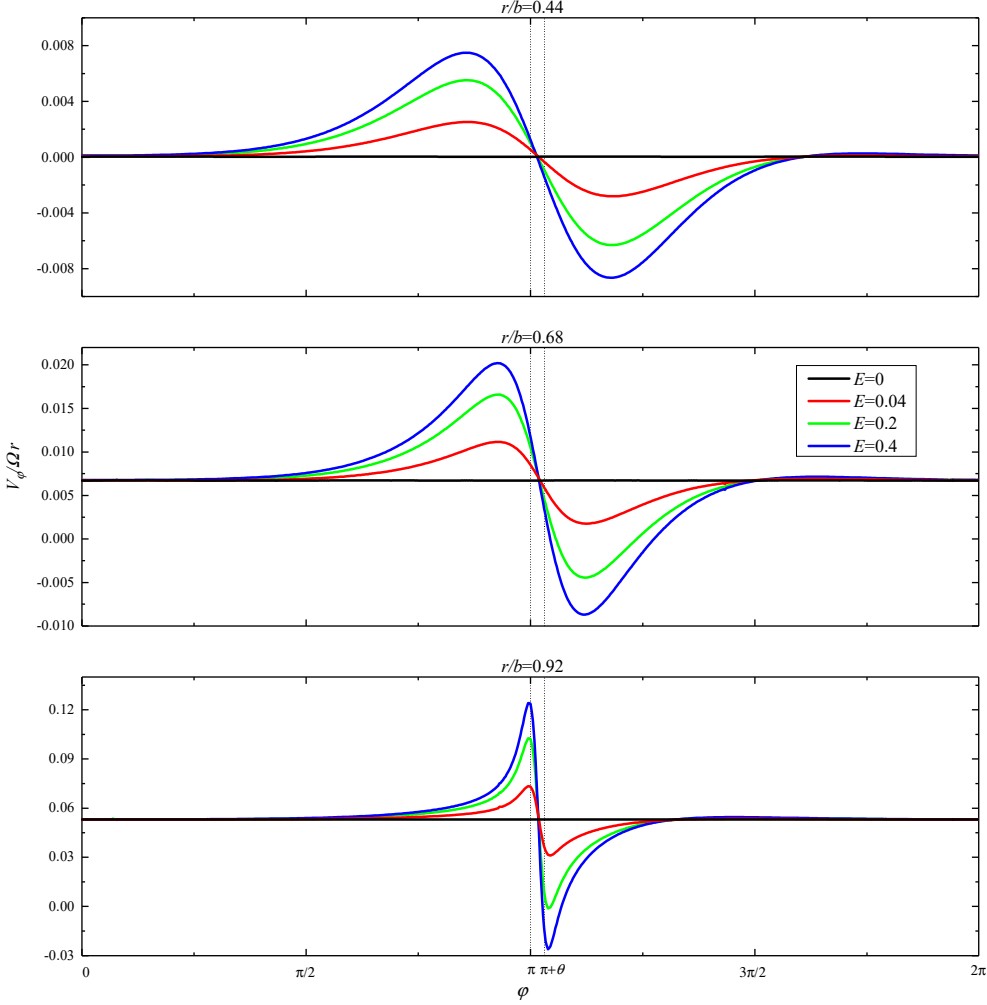

**Figure 8.** $\beta$ curves for $C_w = 10137$, $Re_\varphi = 5.12 \times 10^5$, $\theta = 0.1$ at $z/s = 0.5$.

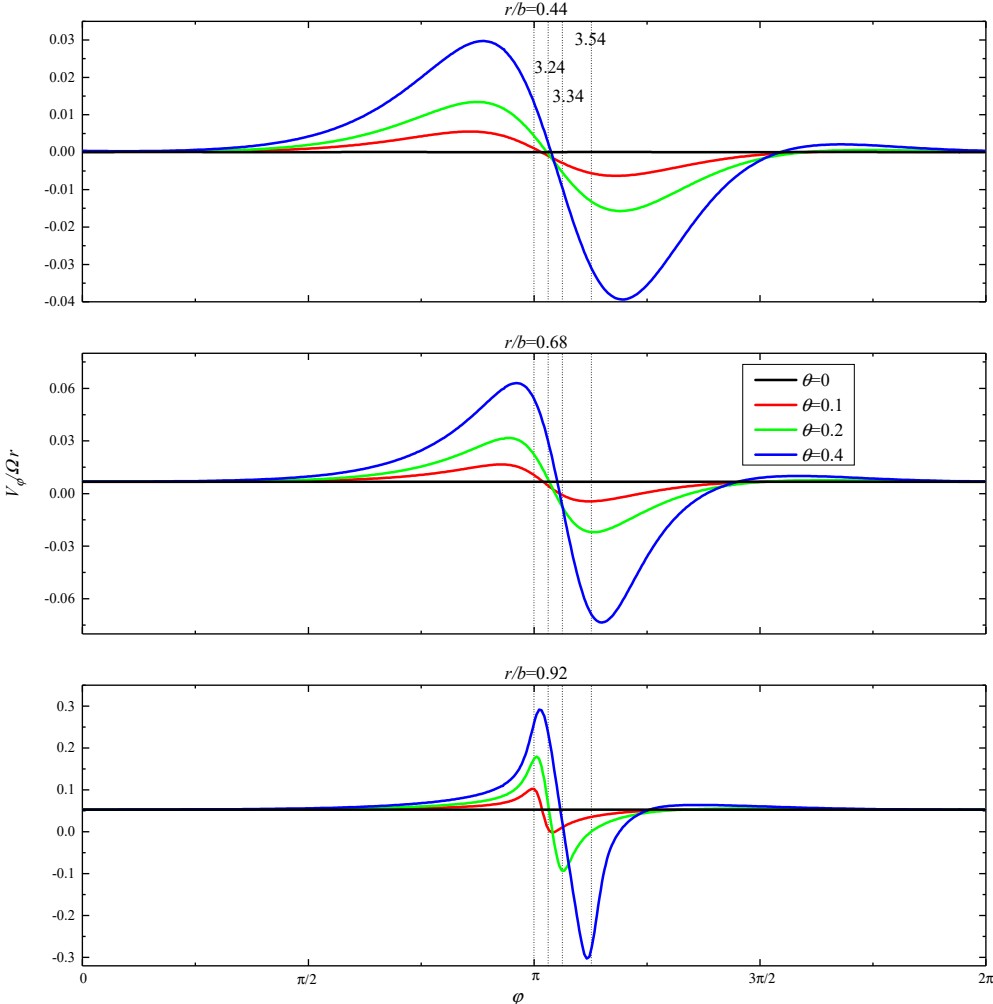

**Figure 9.** $\beta$ curves for $C_w = 10,137$, $Re_\varphi = 5.12 \times 10^5$, $E = 0.2$ at $z/s = 0.5$.

*3.2. Radial Velocity and Mass Flow Rate Distribution*

3.2.1. Radial Velocity

When the turbine blades are not fractured, the flow in the rotor-stator cavity is axisymmetric, and therefore the amount of cold air obtained by each turbine blade is equal. However, when the turbine blade fractures and fails, the flow in the rotor-stator cavity is no longer symmetrical, resulting in an uneven velocity as well as a mass flow rate distribution. This leads to problems with uneven cooling of the turbine disk and the turbine blades. This section will focus on the distribution of radial velocity and mass flow rate after a turbine blade fracture.

The distribution of the dimensionless radial velocity in the circumferential direction for different rotational Reynolds numbers is shown in Figure 10. It is clear that as the $Re_\varphi$ increases, the maximum value at the high-radius position ($r/b = 0.92$) gradually shifts from the middle of the low-pressure zone towards the downstream border. At the same time, the radial velocity distribution becomes more asymmetric and steep, indicating that radial velocity is more sensitive to turbine blade fracture when the $Re_\varphi$ is high. This is also evident from the tangential velocity distribution.

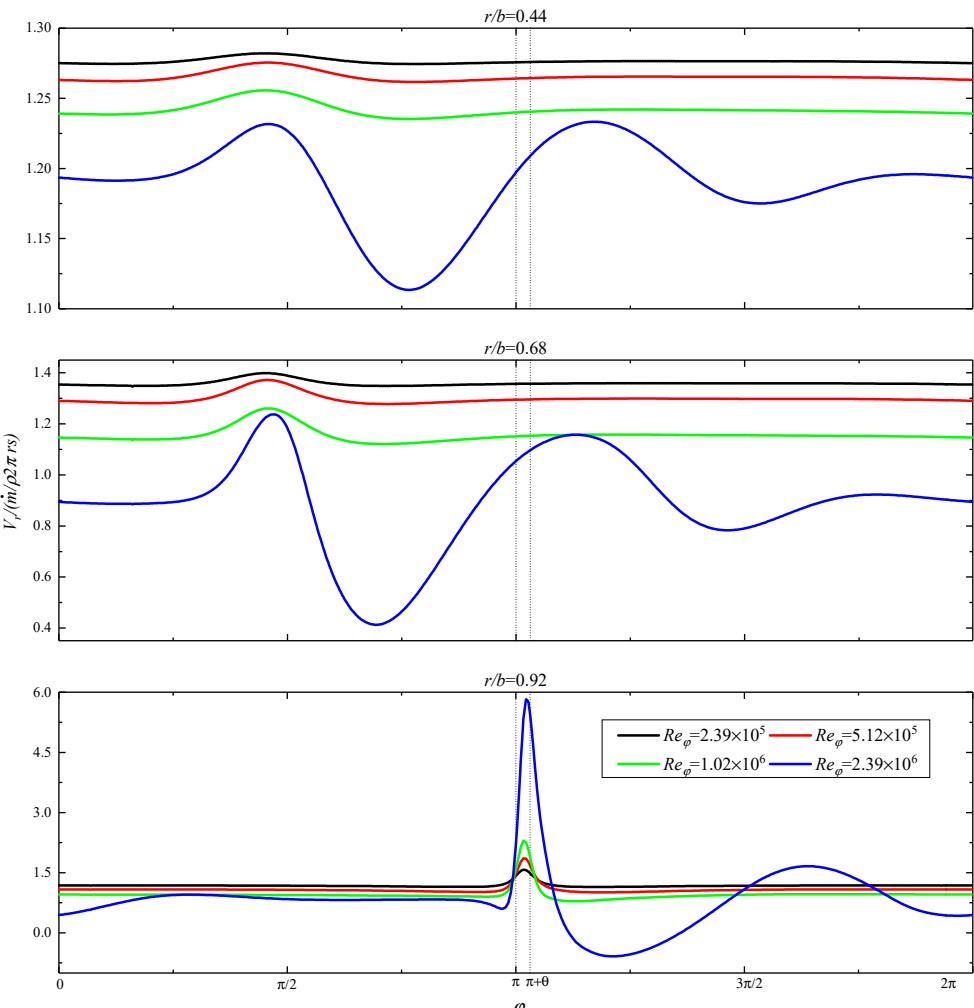

**Figure 10.** Dimensionless radial velocity for $C_w = 10,137$, $E = 0.2$, $\theta = 0.1$ at $z/s = 0.5$.

Figures 11 and 12 show the distribution of the dimensionless radial velocity in the circumferential direction for different $E$ and $\theta$ numbers. As can be seen from the figures, the $E$ and the $\theta$ have a similar effect on the radial velocity distribution. The upstream areas are more affected at the low-radius locations, while the downstream areas are more affected at the high-radius locations. In addition, the high-radius position is more affected than the low-radius position. The standard deviation of the radial velocity at high-radius locations is about one order of magnitude larger than that at low radii. For $E = 0.4$, the standard deviation of the radial velocity at $r/b = 0.44$, 0.68, and 0.92 is 0.00447, 0.02628, and 0.14153, respectively. One phenomenon is worth noting in Figure 11. An increase in the $E$ has little effect on the distribution pattern of the radial velocities, which are essentially symmetrically distributed. For $r/b = 0.92$ and $E = 0.04$, 0.2, and 0.4, the skewness (which can be used to measure the asymmetry of the distribution) is 5.39, 5.70, and 5.84, respectively, while the kurtosis (which can be used to measure the steepness of the distribution) is 32.07, 35.63, and 37.34, respectively.

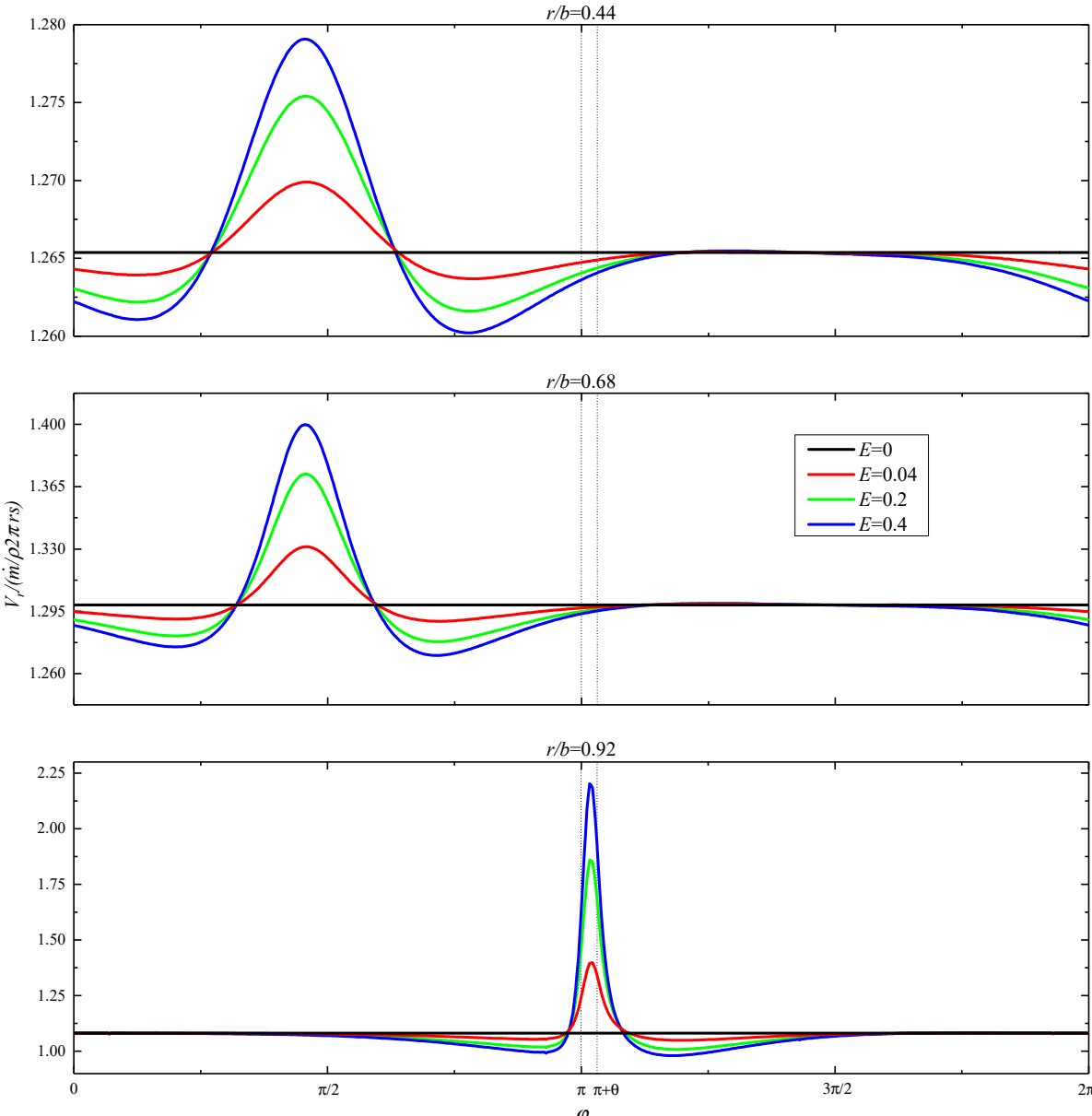

**Figure 11.** Dimensionless radial velocity for $C_w = 10,137$, $Re_\varphi = 5.12 \times 10^5$, $\theta = 0.1$ at $z/s = 0.5$.

As the $\theta$ increases, the distribution of radial velocities becomes increasingly asymmetrical, with the downstream gradually taking the lead in the emergence of recirculation zones (as shown in Figure 12). In addition, the minimum values in the downstream areas decrease by a larger amount relative to $\theta = 0$ than those in the upstream areas. (This phenomenon becomes more pronounced as the $\theta$ increases, as $\theta = 0.4$ and $r/b = 0.92$). This is mainly due to the different magnitudes of the radial coriolis force on the upstream and downstream areas. When $\theta \neq 0$ (turbine blade fracture), the tangential velocities on either side of the low-pressure zone are in opposite directions, and the fluid in the upstream areas is subject to a smaller radial inward coriolis force, while the fluid in the downstream areas is subject to a larger force, so the radial velocity in the downstream areas is smaller and more prone to backflow. This phenomenon is more pronounced at a larger $\theta$, where the difference in tangential velocity between the two sides of the low-pressure zone is greater, and therefore the effect of the coriolis force is more pronounced. This means that when a turbine blade fractures, the downstream area at the high-radius location is more affected, in terms of radial velocity.

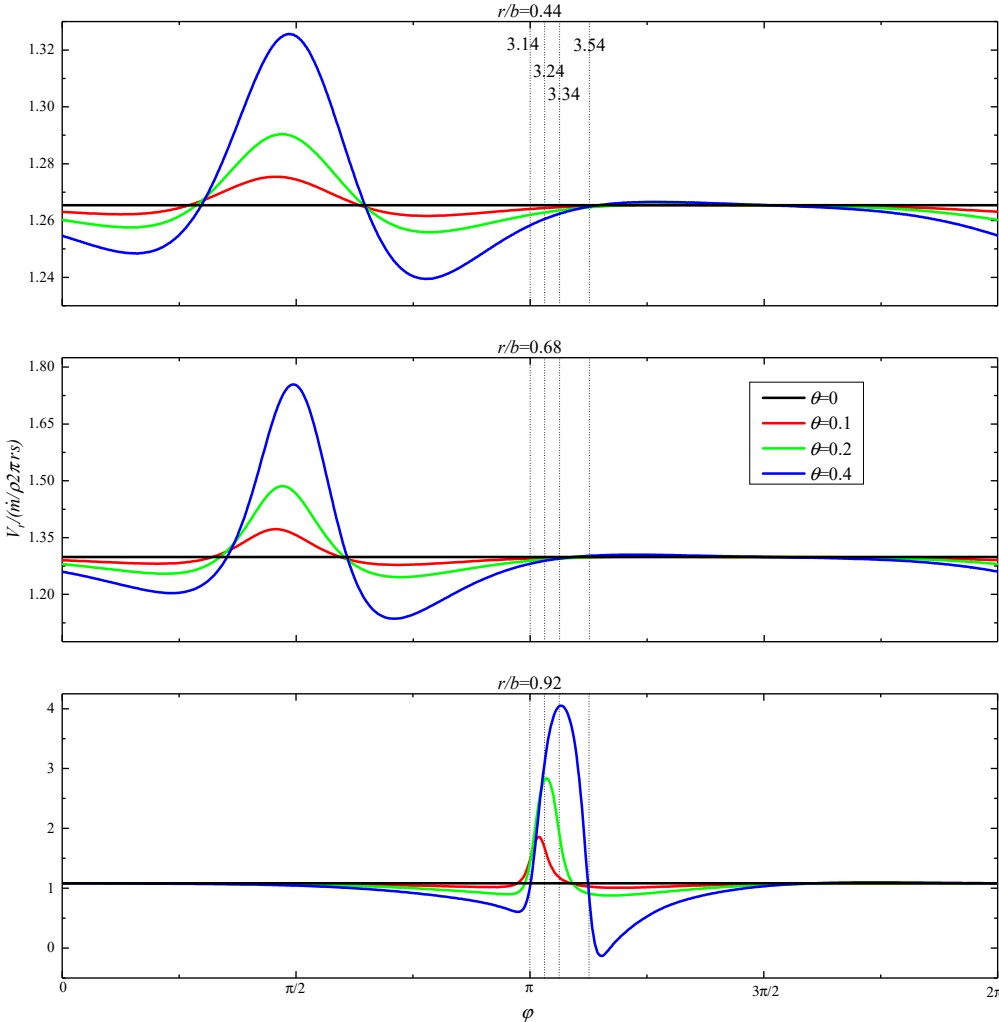

**Figure 12.** Dimensionless radial velocity for $C_w = 10,137$, $Re_\varphi = 5.12 \times 10^5$, $E = 0.2$ at $z/s = 0.5$.

3.2.2. Mass Flow Rate

Figure 13 shows the distribution of the mass flow rate in the circumferential direction at different $Re_\varphi$ values with the vertical coordinates indicating the ratio of the actual mass flow rate to the average mass flow rate. It can be seen from the figure that as the $Re_\varphi$ increases, backflow gradually appears in the area near the upstream and downstream borders, and the outflow in the low-pressure area also increases significantly due to mass flow rate conservation. At $Re_\varphi = 2.39 \times 10^6$, the maximum mass flow rate is 27 times larger than the average value. Meanwhile, as the $Re_\varphi$ increases, the location of the maximum mass flow rate gradually moves from the middle of the low-pressure zone towards the downstream border. Another point of interest is that at the current condition ($\theta = 0.1$), the mass flow rate at the upstream border is always smaller than that at the downstream border, regardless of how the $Re_\varphi$ varies. (This phenomenon is of course more pronounced when the $Re_\varphi$ is larger). This is because the fluid at the low-pressure zone is subject to a higher counter-rotating tangential coriolis force, due to the higher radial velocity than that at other areas, which pushes the low-pressure fluid in the low-pressure zone upstream, eventually leading to a large inverse pressure at the upstream border (as shown in Figure 14, the pressure coefficient is defined by Equation (4)). Therefore, this pressure is more likely to produce backflow upstream.

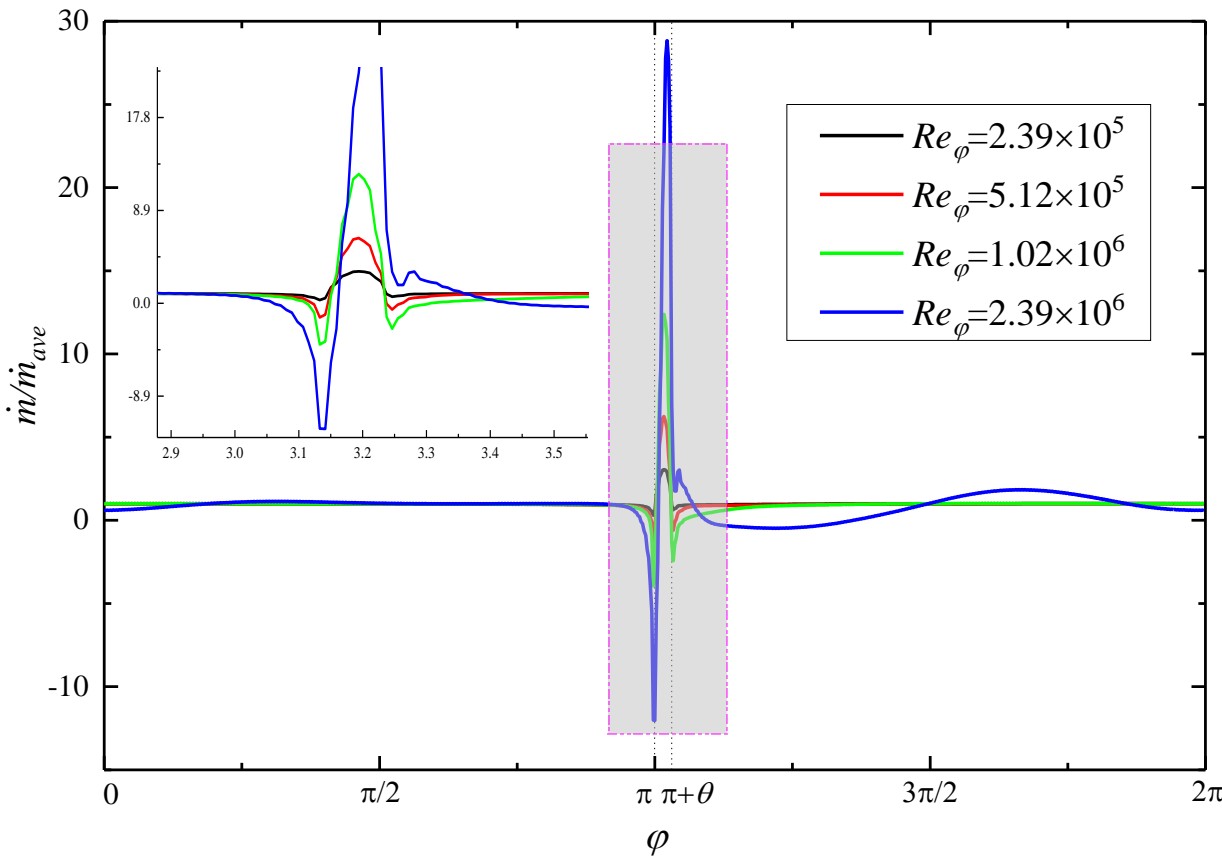

**Figure 13.** Distribution of mass flow rate for $C_w = 10,137$, $E = 0.2$, $\theta = 0.1$ at $r/b = 1$.

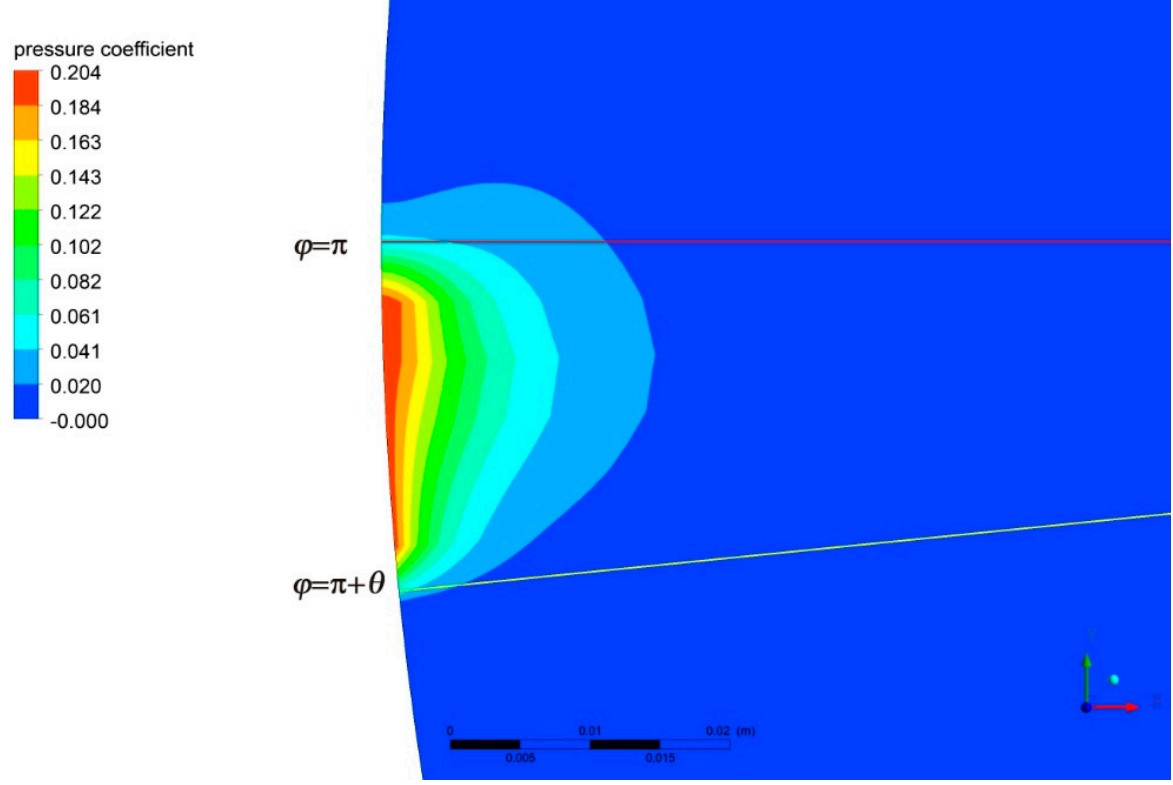

**Figure 14.** Pressure coefficient of case D at plane $z/s = 0.5$.

In addition, as the $Re_\varphi$ increases, the radial inward coriolis force on the downstream region gradually increases (i.e., the tangential velocity relative to the rotor increases, as does the rotational speed, as shown in Figure 5), so backflow also begins to occur. From the above analysis, it is clear that when a particular blade fractures ($\theta = 0.1$), the flow rate of its upstream blade will be reduced or even experience gas intrusion, and the blade is therefore more likely to fracture; continuing to increase the $Re_\varphi$ will also affect the downstream blade.

Figure 15 shows the distribution of the mass flow rate in the circumferential direction for different Euler numbers. As in the analysis above, the variation in the *E* does not affect the distribution pattern of the mass flow rate; the mass flow rate remains maximum in the middle of the low-pressure zone and minimum at the upstream and downstream borders. As the *E* increases, the maximum and minimum values increase and decrease, respectively, and the mass flow rate at the upstream border is always smaller than that at the downstream border. The reasons for this are analyzed above.

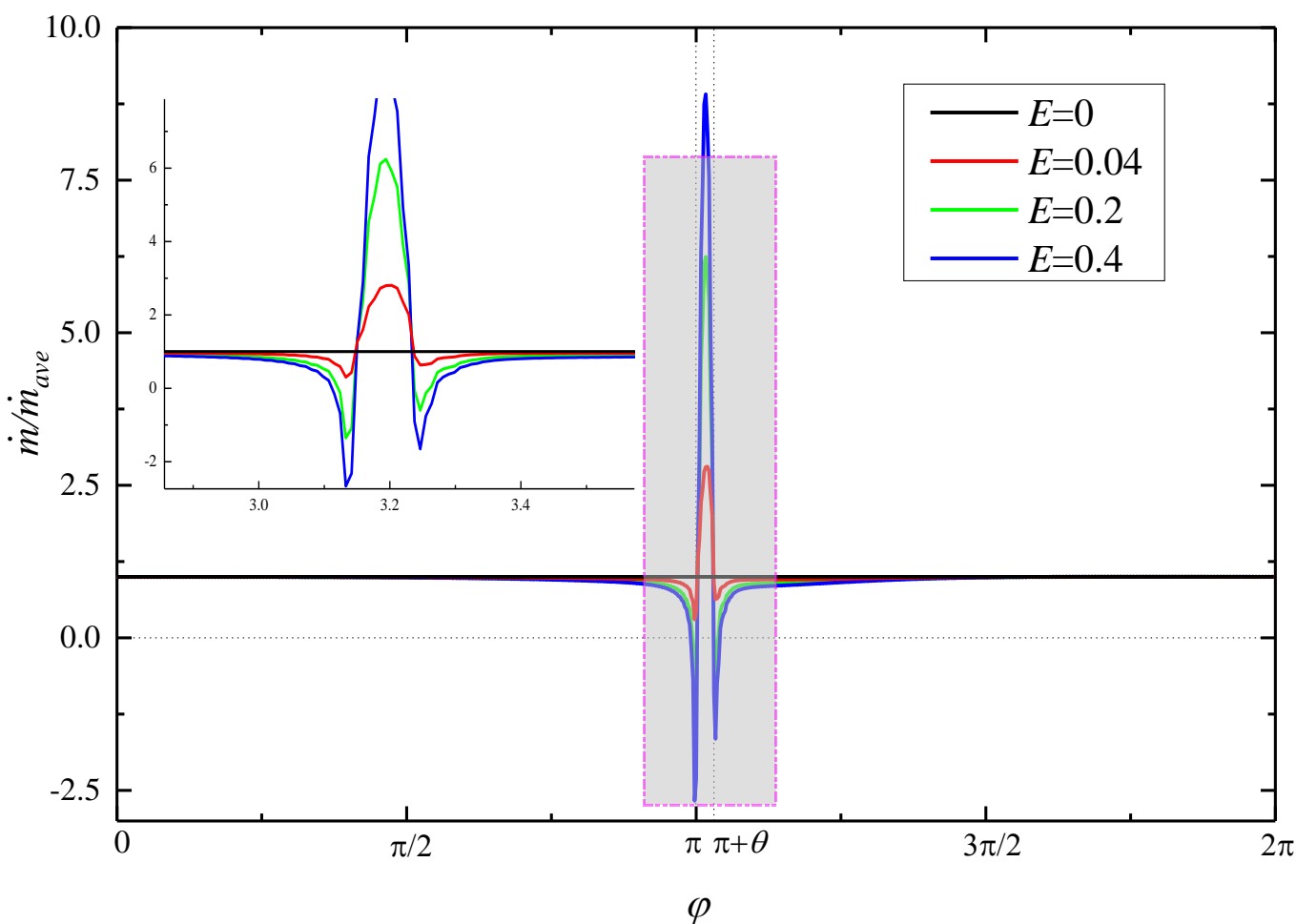

**Figure 15.** Distribution of mass flow rate for $C_w = 10,137$, $Re_\varphi = 5.12 \times 10^5$, $\theta = 0.1$ at $r/b = 1$.

Figure 16 shows the distribution of the mass flow rate in the circumferential direction for different $\theta$ values It is clear from the figure that the maximum values of the mass flow rate occur in the middle of the low-pressure zone when $Re_\varphi = 5.12 \times 10^5$, regardless of the variation of the $\theta$, and that the maximum values are almost equal. When the $\theta$ is small ($\theta = 0.1$), backflow occurs at both the upstream and downstream borders, and the mass flow rate at the upstream border is smaller than that at the downstream border; as the $\theta$ increases ($\theta = 0.2$), the backflow zone disappears and the mass flow rate at the upstream and downstream borders is almost equal; upon continuing to increase the $\theta$ ($\theta = 0.4$),

the backflow zone reappears, but the mass flow rate at the downstream border is smaller than that at the upstream border. It seems that when $\theta = \theta_c$ ($\theta_c = 0.2$, when $C_w = 10,137$, $Re_\varphi = 5.12 \times 10^5$, and $E = 0.2$), the backflow zone disappears and the mass flow rate of the upstream and downstream border are equal; when $\theta > \theta_c$, the backflow zone gradually appears first at the downstream border; and when $\theta < \theta_c$, the backflow zone gradually appears first at the upstream border.

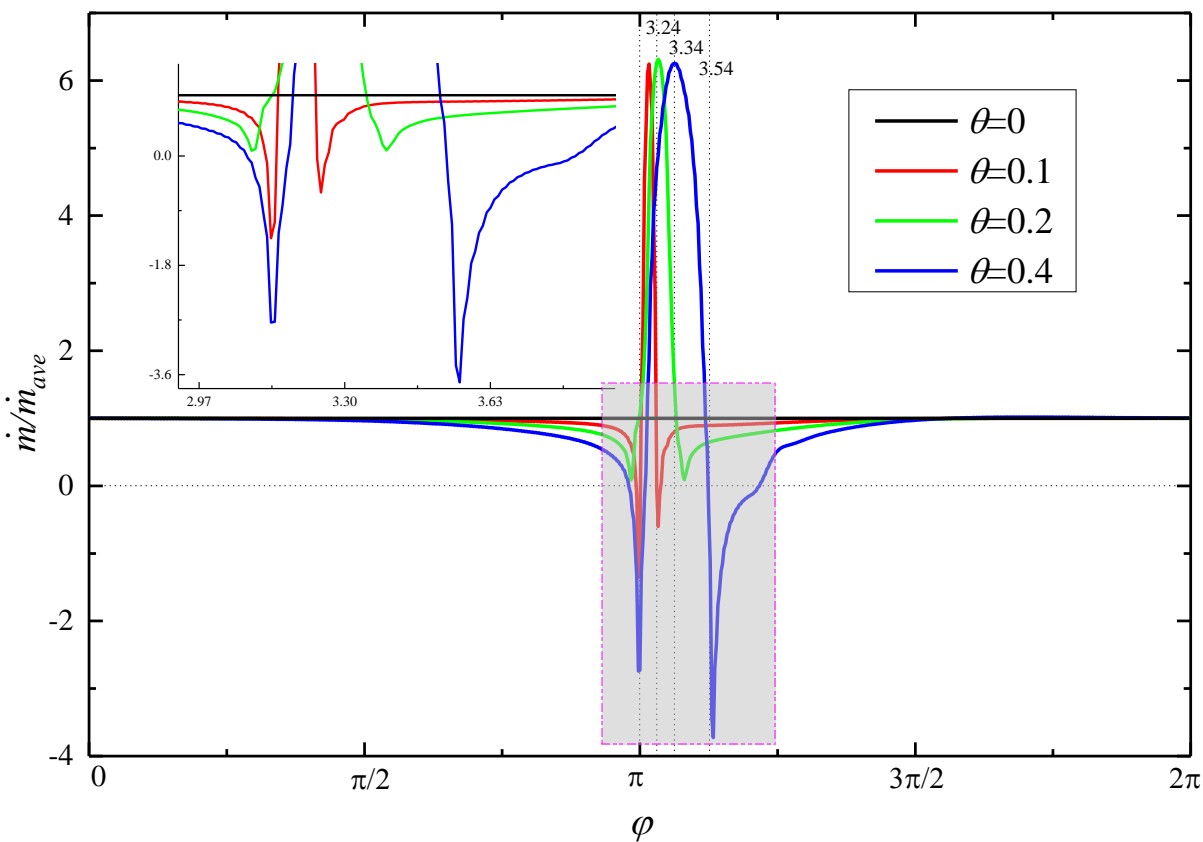

**Figure 16.** Distribution of mass flow rate for $C_w = 10,137$, $Re_\varphi = 5.12 \times 10^5$, $E = 0.2$ at $r/b = 1$.

The reason for this phenomenon is that when the $\theta$ is small, the tangential velocity is small, the radial coriolis forces have little influence, and the pressure dominates, while the inverse pressure gradient near the upstream border is larger (see Figure 14), so the mass flow rate near the upstream border is smaller. As the $\theta$ increases, the radial coriolis force gradually increases, but the radial inward coriolis force near the downstream border is larger than at the upstream border, superimposed on the influence of pressure, until $\theta = \theta_c$, when the upstream and downstream borders are subject to the same radial combined force. In this case, the mass flow rate of upstream and downstream borders is equal. Upon continuing to increase the $\theta$, the effect of the coriolis force is greater, while the Euler number remains the same (the effect of pressure remains the same), so the downstream border is subject to a greater radial inward force, resulting in a smaller mass flow rate at the downstream border than at the upstream border. For the cooling of turbine blades, it is vital to find the $\theta_c$. If a blade fractures at a point where $\theta = \theta_c$, then the distribution of cold air will be as balanced as possible, thus potentially avoiding the serious consequences of gas intrusion and ensuring the safety of the engine.

From the above analysis, it is clear that the $\theta_c$ will increase as the $Re_\varphi$ and the $E$ increase. Figure 17 shows the distribution of the mass flow rate in the circumferential direction for different $\theta$ when $C_w = 10,137$, $Re_\varphi = 5.12 \times 10^5$, and $E = 0.4$. The figure shows that for $\theta = 0.2$, the flow rate at the upstream border is still smaller than that at downstream border, so the value of the $\theta_c$ should be slightly greater than 0.2.

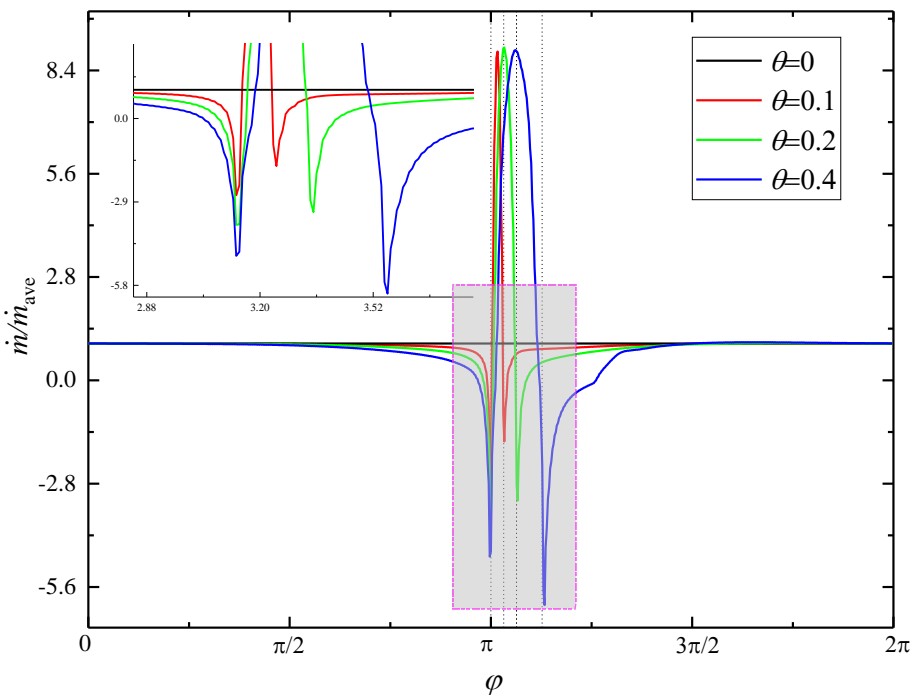

**Figure 17.** Distribution of mass flow rate for $C_w = 10,137$, $Re_\varphi = 5.12 \times 10^5$, $E = 0.4$ at $r/b = 1$.

### 3.3. Pressure and Thrust

#### 3.3.1. Pressure Coefficient

The axial thrust of a gas turbine is critical to the performance and safety of the engine. However, to the authors' knowledge, there are no studies to date on the change in turbine disk axial thrust after turbine blade fracture. This section investigates the distribution of pressure and then analyses the variation of axial thrust. In previous studies, Will et al. [2] used Equation (1) to evaluate the pressure distribution along the radius of the disk in a rotor-stator cavity with through-flow for an incompressible, steady flow.

$$\frac{\partial p}{\partial r} = \rho \beta C_{qr}{}^2 \Omega^2 r + \frac{\rho Q^2}{4\pi^2 s^2 r^3} \tag{1}$$

Based on Equation (1), the pressure along the radius of the disk can be calculated with Equation (2) by Hu et al. [24].

$$p(r) = p_b + \int_b^r \rho \beta C_{qr}{}^2 \Omega^2 r dr + \frac{\rho Q^2}{8\pi^2 s^2} \left( \frac{1}{b^2} - \frac{1}{r^2} \right) \tag{2}$$

Dimensionless pressure is defined as follows:

$$P^* = \frac{p}{\rho \Omega^2 b^2} \tag{3}$$

The pressure coefficient is defined as:

$$C_p = P^* \left( \frac{r}{b} = 1 \right) - P^* \left( \frac{r}{b} \right) \tag{4}$$

Figure 18 shows the pressure coefficient distribution for the different circumferential positions of cases B, D, F, and H, numbered a, b, c and d, respectively. As can be seen from Figure 18a, the pressure coefficients obtained by Equation (2) are in very good agreement with the SST model, indicating that the SST model is suitable for this problem. Throughout these four plots, it can be seen that the pressure coefficients at the low-radius locations are almost unaffected, in line with the previous analysis. In contrast, at high radii, the pressure

coefficients are generally larger at the middle of the low-pressure zone ($\phi = \pi + \theta/2$) than at other locations (except in case H), and larger at the downstream border ($\phi = \pi + \theta$) than at the upstream border ($\phi = \pi$). Comparing the plots of cases B and D, the pressure coefficients of upstream and downstream borders tend to be equal at high-radius locations as the $Re_\varphi$ increases. Comparing the two plots of cases B and F, a change in the Euler number does not significantly affect the distribution pattern of the pressure coefficients; only the magnitude of the values changes. Comparing the graphs of cases B and H, the pressure coefficient at the downstream border even exceeds that at the middle of the low-pressure zone as the $\theta$ increases. This corresponds to the fact that the downstream border is the first to experience backflow as the $\theta$ increases (see Figure 16).

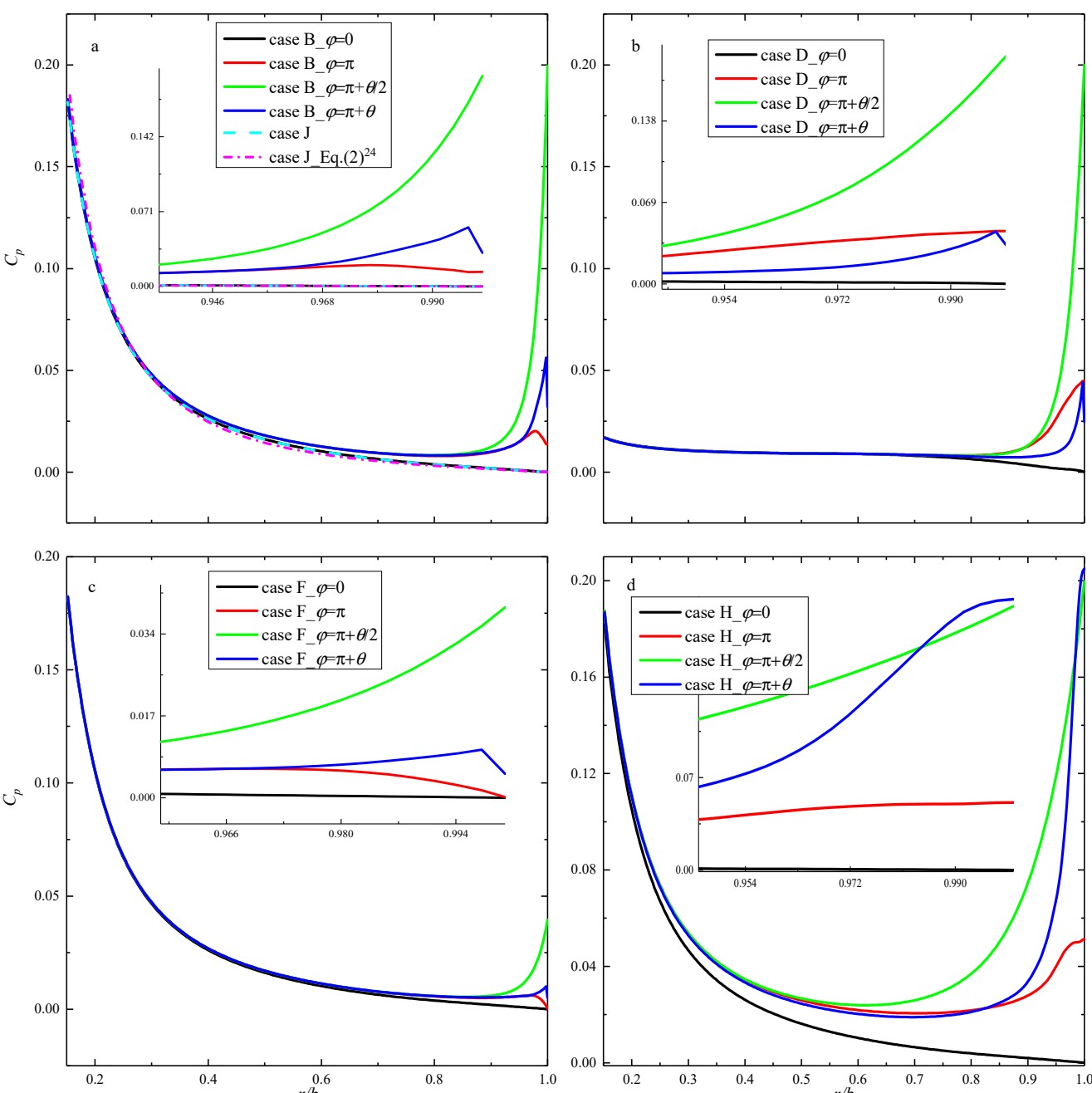

**Figure 18.** Pressure coefficient of different cases at plane $z/s = 0.5$: (**a**) case B, (**b**) case D, (**c**) case F, (**d**) case H.

Figure 19 shows the distribution of pressure coefficients at different $Re_\varphi$ values, where a, b, c and d represent different circumferential positions. As the $Re_\varphi$ increases, the pressure coefficients at each location do not vary much, except for a slight increase near the upstream border at the high-radius location. The pressure coefficient decreases as the $Re_\varphi$ increases. However, by continuing to increase the $Re_\varphi$, the pressure coefficient at high radii increases (see the enlargement of Figure 19a). Furthermore, there is a sudden drop in the pressure coefficient at the downstream border ($\phi = \pi + \theta$) near the outlet. This is mainly because the fluid in the low-pressure zone is carried upstream and the downstream is filled with high-pressure fluid (similar to Figure 14).

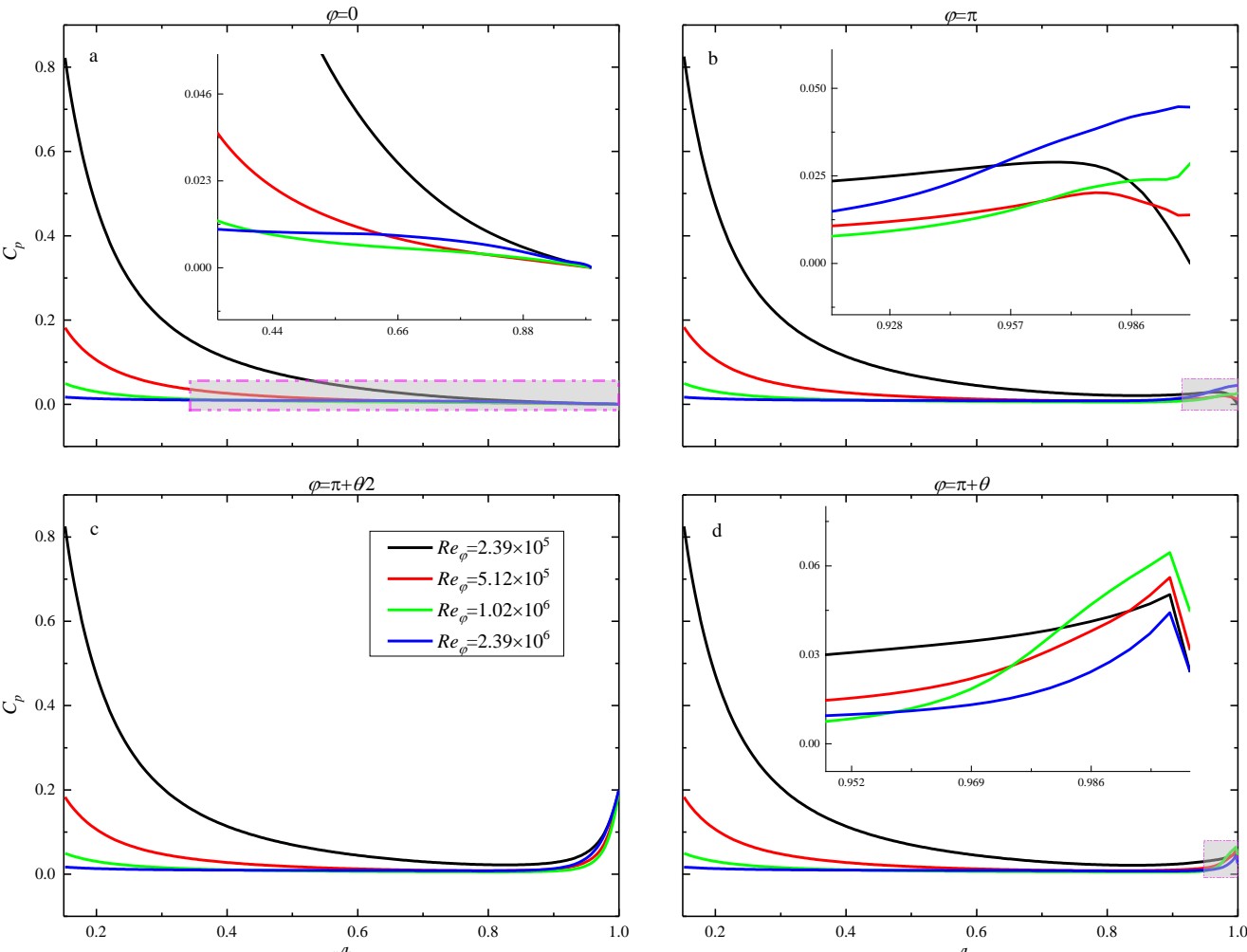

**Figure 19.** Pressure coefficient for $C_w = 10,137$, $E = 0.2$, $\theta = 0.1$ at plane $z/s = 0.5$: (**a**) $\phi = 0$, (**b**) $\phi = \pi$, (**c**) $\phi = \pi + \theta/2$, (**d**) $\phi = \pi + \theta$.

Figure 20 shows the distribution of the pressure coefficients for different Euler numbers, where a, b, c and d represent different circumferential positions. It is clear that an increase in the Euler number does not affect the pressure at low radii. As for the higher radii, the pressure coefficients in and around the low-pressure zone increase significantly as the Euler number increases.

The distribution of pressure coefficients for different $\theta$ values is shown in Figure 21, where a, b, c and d represent different circumferential positions. Unlike the $E$ and the $Re_\varphi$, variations in the $\theta$ can affect the pressure coefficient even at low-radius locations, resulting in an increase in the pressure coefficient in the low-pressure region and its immediate vicinity.

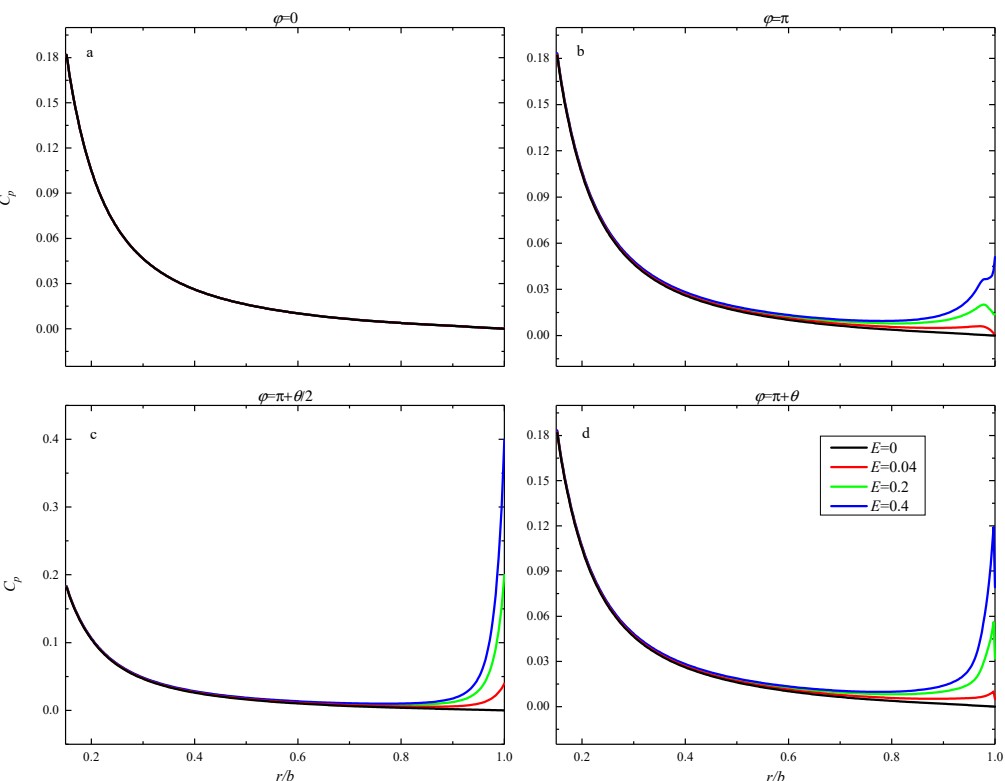

**Figure 20.** Pressure coefficient for $C_w = 10,137$, $Re_\varphi = 5.12 \times 10^5$, $\theta = 0.1$, at plane $z/s = 0.5$: (**a**) $\phi = 0$, (**b**) $\phi = \pi$, (**c**) $\phi = \pi + \theta/2$, (**d**) $\phi = \pi + \theta$.

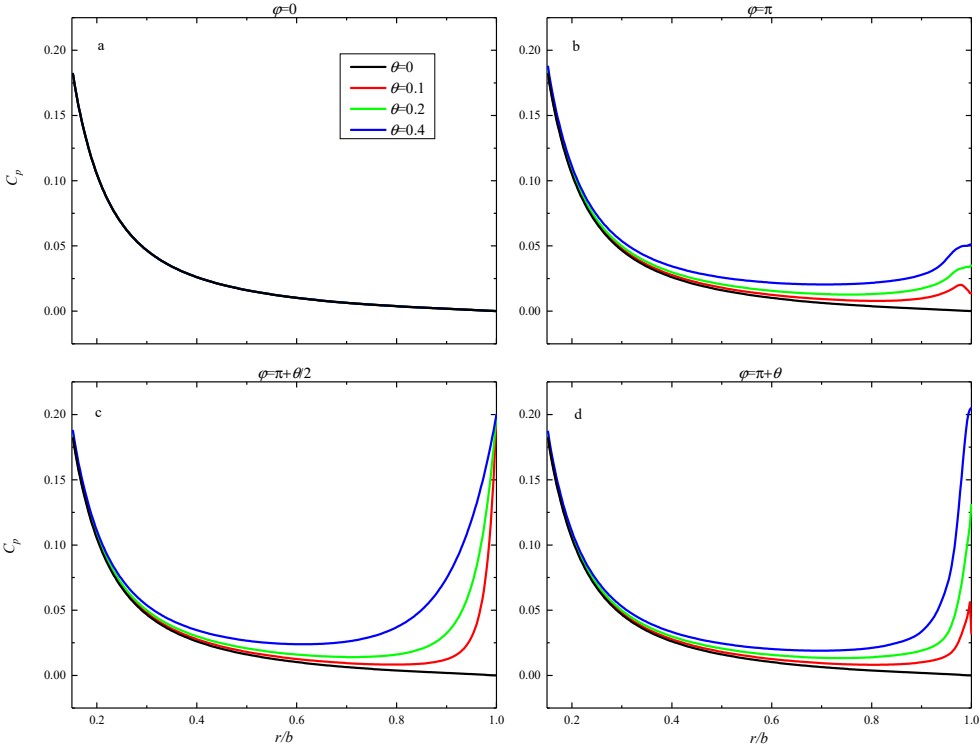

**Figure 21.** Pressure coefficient for $C_w = 10,137$, $Re_\varphi = 5.12 \times 10^5$, $E = 0.2$, at plane $z/s = 0.5$: (**a**) $\phi = 0$, (**b**) $\phi = \pi$, (**c**) $\phi = \pi + \theta/2$, (**d**) $\phi = \pi + \theta$.

### 3.3.2. Thrust Coefficient

Figure 22 shows the thrust coefficients of the rotor and stator for different cases, where a, b and c show the impact of the $Re_\varphi$, E and $\theta$ respectively. The thrust coefficient characterizes the ratio of the axial force on the disk to the centrifugal force, which is defined as in [25]:

$$C_F = \frac{\iint (p_b - p)ds}{\rho \Omega^2 b^4} = \int_a^b \frac{2\pi(p_b - p)rdr}{\rho \Omega^2 b^4} \tag{5}$$

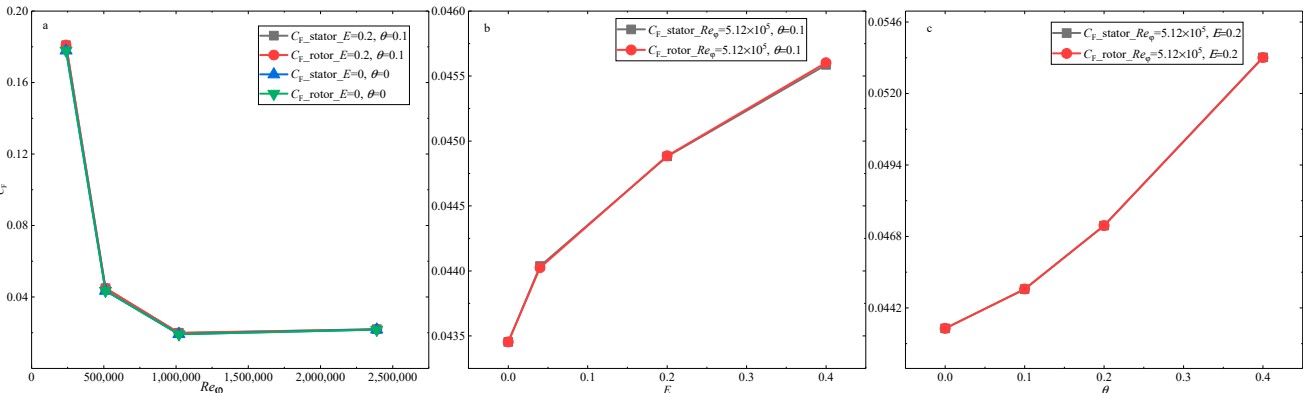

**Figure 22.** Thrust coefficient of rotor and stator: (**a**) $E = 0.2$, $\theta = 0.1$; (**b**) $Re_\varphi = 5.12 \times 10^5$, $\theta = 0.1$; (**c**) $Re_\varphi = 5.12 \times 10^5$, $E = 0.2$.

By definition, $C_F$ and $C_p$ are related as follows:

$$C_F = \frac{\iint (p_b - p)ds}{\rho \Omega^2 b^4} = \frac{1}{b^2} \iint \frac{(p_b - p)ds}{\rho \Omega^2 b^2} = \frac{1}{b^2} \iint C_p ds \tag{6}$$

It is clear that the thrust coefficients of the rotor and the stator are almost equal. $E = 0$ and $\theta = 0$ mean the blades are intact, while $E = 0.2$ and $\theta = 0.1$ means a blade is fractured. From Figure 22a, it can be seen that the disk thrust coefficient of the stator increases by 1.7%, 3.3%, 3.8%, and 0.8%, when the $Re_\varphi = 2.39 \times 10^5$, $5.12 \times 10^5$, $1.02 \times 10^6$, and $2.39 \times 10^6$, respectively, because of blade fracture. This is mainly due to the fracture of the turbine blades, resulting in a low-pressure area on the disk surface. In addition, the increased ratio first increases and then decreases as the $Re_\varphi$ increase. When the blades are already fractured, the thrust coefficient of the disks decreases significantly as the $Re_\varphi$ increases but as the $Re_\varphi$ continues to increase, the thrust coefficient increases. This is the result of the combined effect of centrifugal force and the low-pressure zone. When the $Re_\varphi$ is small, increasing the $Re_\varphi$ will increase the effect of centrifugal force, making the pressure coefficient gradually converge to zero across the disk (refer to Figure 19), so the thrust coefficient decreases significantly. However, by continuing to increase the $Re_\varphi$, the pressure coefficient at high radii increases (see the enlargement of Figure 19a). Therefore, continuing to increase the $Re_\varphi$ will increase the thrust coefficient slightly, according to Equation (6). This also means that the thrust coefficient is more sensitive to the $Re_\varphi$ whether blades fracture or not, especially when the $Re_\varphi$ is small. Figure 22b shows that an increase in the Euler number increases the thrust coefficient. Figure 22c shows that an increase in the $\theta$ also increases the thrust coefficient, and the effect of the $\theta$ on the thrust coefficient is approximately linear.

### 3.4. Moment Coefficient

Figure 23 shows the moment coefficients of the rotor and the stator on the rotor shaft $z$ for different cases, where a, b and c show the impact of the $Re_\varphi$, E and $\theta$ respectively. The moment coefficient is defined by Equation (7) [25]:

$$C_M = \frac{2 \cdot |M|}{\rho \Omega^2 b^5} \tag{7}$$

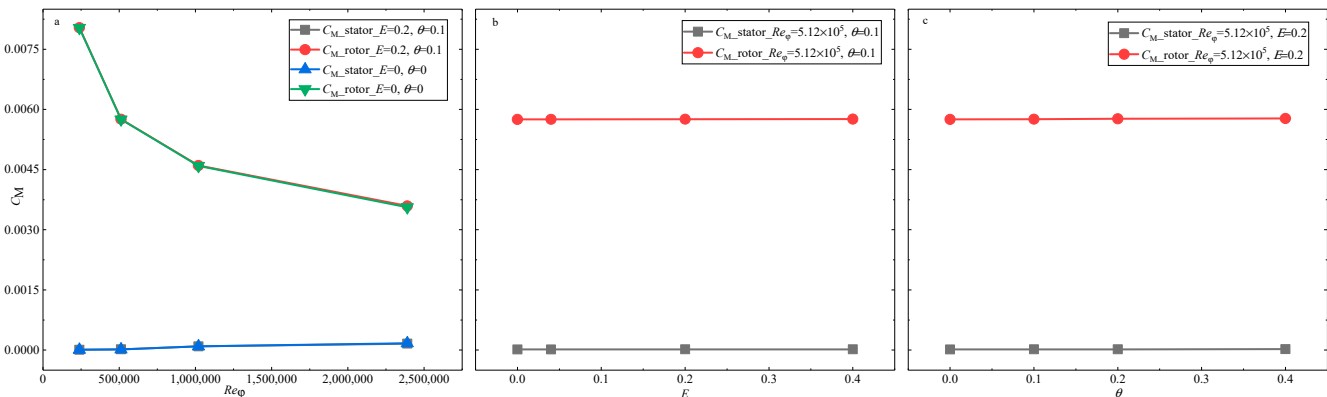

**Figure 23.** Moment coefficient of rotor and stator on the rotating shaft (z): (**a**) $E = 0.2$, $\theta = 0.1$; (**b**) $Re_\varphi = 5.12 \times 10^5$, $\theta = 0.1$; (**c**) $Re_\varphi = 5.12 \times 10^5$, $E = 0.2$.

From Figure 23, it can be seen that the moment coefficient of the rotor is much greater than that of the stator. In addition, the $E$ (turbine blade fracture) and the $\theta$ have almost no effect on the moment coefficient. Figure 23a shows that as the $Re_\varphi$ increases, the moment coefficient of the rotor decreases while that of the stator increases, whether or not blades fracture. This is mainly because as the $Re_\varphi$ increases, the swirl ratio increases, and therefore the moment coefficient of the rotor decreases (according to Han et al. [26]). However, an increase in the swirl ratio increases the relative velocity of the fluid to the stator, and therefore the moment coefficient of the stator increases.

After the turbine blade fracture, a low-pressure zone appears and the pressure distribution in the rotor-stator cavity is no longer symmetrical, so the moment coefficient on the radial direction is not zero. This means that the turbo disk tends to roll over on its side. The moment coefficients for the rotor and the stator on $\varphi = \pi/2 + \theta/2$ are shown in Figure 24, where a, b and c show the impact of the $Re_\varphi$, E and $\theta$ respectively. It is equal in magnitude to that on the rotating shaft. The moment coefficient in this direction is the largest of all radial directions. The moment coefficients of the stator and the rotor are essentially the same. Figure 24a shows that as the $Re_\varphi$ increases, the moment coefficient decreases. This is because the pressure coefficient gradually converges to zero along the radius as the $Re_\varphi$ increases. From Figure 24b,c, it can be seen that increasing the $E$ and the $\theta$ increases the moment coefficient.

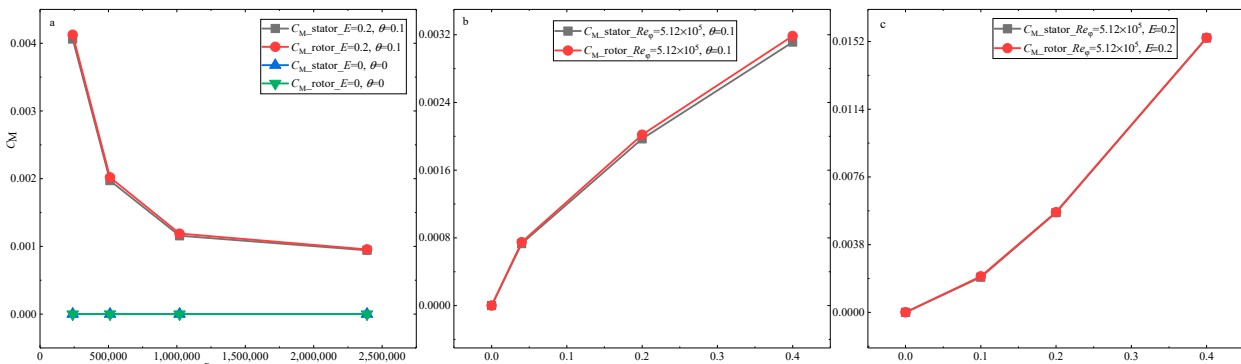

**Figure 24.** Moment coefficient of rotor and stator on the $\varphi = \pi/2 + \theta/2$: (**a**) $E = 0.2$, $\theta = 0.1$; (**b**) $Re_\varphi = 5.12 \times 10^5$, $\theta = 0.1$; (**c**) $Re_\varphi = 5.12 \times 10^5$, $E = 0.2$.

## 4. Conclusions

Turbine blade fracture can lead to engine safety problems, so this paper investigated the effect of turbine blade fracture on the flow in a simple rotor-stator system without a shroud, employing numerical simulation. The results of the numerical simulations were compared with relevant empirical correlations. Specifically, this paper studied the effects of control parameters, such as the rotational Reynolds number, the Euler number, and the range of low-pressure zones on the velocity and pressure fields, as well as the thrust coefficient, the moment coefficient, and heat transfer in a simple rotor-stator cavity. Within the scope of this paper's research, the following conclusions can be drawn.

1. For the swirl ratio, the effects of the rotational Reynolds number, the Euler number, and the $\theta$ are similar. In addition, although the downstream region is more affected than the upstream region, an increase in the Euler number and the $\theta$ increases the swirl ratio variation, while an increase in the rotational Reynolds number decreases the swirl ratio variation.

2. Increases in the rotational Reynolds number, the Euler number, and the $\theta$ all lead to a more uneven distribution of the flow rate. Furthermore, regardless of the rotational Reynolds number and the Euler number, the flow rate at the upstream border is always smaller than at the downstream border, but an increase in the $\theta$ may lead to a more balanced flow rate distribution (there is a critical $\theta_c$ that makes the flow rate distribution most balanced; $\theta_c \cong 0.2$ when $C_w = 10,137$, $Re_\varphi = 5.12 \times 10^5$, and $0.2 \leq E \leq 0.4$).

3. Turbine blade fracture causes an increase in the thrust coefficient and is more pronounced at smaller rotational Reynolds numbers. The increase in the thrust coefficient does not exceed 4% when $E = 0.2$, $\theta = 0.1$, as discussed in this paper.

4. Changes in the rotational Reynolds number, the Euler number, and the $\theta$ have almost no effect on the moment coefficient about the axis of rotation but have a more significant effect on the moment coefficient about the radial direction. The latter will decrease as the rotational Reynolds number increases and increase as the Euler number and the $\theta$ increase.

**Author Contributions:** Funding acquisition, T.Q. and P.L.; writing–original draft, G.Z. All authors have read and agreed to the published version of the manuscript.

**Funding:** This research was funded by Project of National Science Foundation of China, grant number 61890923. The APC was funded by Project of National Science Foundation of China No. 61890923.

**Data Availability Statement:** Not applicable.

**Acknowledgments:** The authors express their sincere gratitude to the support of the National Fund.

**Conflicts of Interest:** The authors declare no conflict of interest.

## Abbreviations

| | |
|---|---|
| $a$ | inlet radius, m |
| $B$ | radius of rotor and stator, m |
| $S$ | axial spacing between rotor and stator, m |
| $A_c$ | effective sealing area, m$^2$ |
| $\delta A_e, \delta A_i$ | area of a small orifice where air egresses/ingresses, m$^2$ |
| $C_w$ | dimensionless mass flow rate, $m/\mu b$ |
| $E$ | Euler number, $(P_2 - P_1)/0.5\,\rho\Omega^2 b^2$ |
| $G$ | gap ratio, s/b |
| $\dot{m}$ | mass flow rate, kg/s |
| $N$ | number of blades |
| $P_1, P_2$ | outlet pressure of rotor-stator cavity, Pa |
| $P^*$ | dimensionless pressure difference, $(P_2 - p)/0.5\rho\Omega^2 r^2$ |
| $P(\theta)$ | pressure profile of outlet, Pa |
| $Q$ | volume flow rate, m$^3$/s |
| $r^*$ | dimensionless radius, r/b |
| $Re_s$ | rotational Reynolds number based on s, $\Omega s^2/\nu$ |
| $Re_r$ | radial to rotational Reynolds number, $Re_r = C_w/2\pi G Re_\varphi$ |
| $Re_\varphi$ | rotational Reynolds number based on b, $\Omega b^2/\nu$ |
| $V_r, V_\varphi$ | radial and tangential velocity, m/s |

**Greek**

| | |
|---|---|
| $\beta$ | swirl ratio, $V_\varphi/\Omega r$ |
| $\theta$ | the range of low-pressure area, rad |
| $\Omega$ | rotating velocity, rad/s |
| $\nu$ | kinematic viscosity, m$^2$/s |
| $\rho$ | density, kg/m$^3$ |
| $\lambda_T$ | turbulent flow parameter, $C_w/Re_\varphi^{0.8}$ |

**Subscript**

| | |
|---|---|
| $r, z, \phi$ | radial, axial, and tangential coordinates, m/m/rad |
| $c$ | critical |
| 1, 2 | different location |

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
