# Peer review of "Influence of Blade Fracture on the Flow of Rotor-Stator Systems with Centrifugal Superposed Flow"

_aerospace, doi:10.3390/aerospace9020106_

Round 1

Reviewer 1 Report

The paper presents a relatively simple method for the numerical simulation of the stator-rotor cavity flow in radial turbines with failure of some blades. While emphasis is on modelling of such blades by a pure decrease of exit cavity pressure, Sec. 1 and 2 do not explain with enough clarity how the simplified computational domain (basically an annulus) is obtained from the original geometry e.g. the location of the real blades or the interfaces between cavity and main flows is never addressed nor shown. The authors also claim the novelty of a CFD-based approach over previous analytical or linearized methods. However, they adopt an incompressible flow model with apparently specifed massflow at the cavity inlet so no thermal effect can be taken into account. The computational method needs to be addressed more in details. Finally, the title and abstract do not state that the method only concerns radial inflow turbines, and if this is not the case, they should clarify since the beginning which types of machine cavities can be investigated.

In the reviewer's opinion, the above issues need to be resolved in order for the paper to reach Aerospace archival quality.     

Reviewer 2 Report

The paper studies the impact of blade-off on the flow in the stator-rotor cavity. It is assumed that the case in mind is with stator upstream and rotor downstream of the cavity. The loss of a turbine blade would then result in the lack of the potential effect of that particular blade, and a local low pressure zone in that region. If this is the case considered, it may be mentioned in the paper. 

Please mention if any averaging was used when producing the results in the paper.

Were any large pressure structures encountered in the simulations making averaging necessary?

The paper is uploaded with comments for the authors.

The paper was interesting, only requiring small adjustments. If the results are further tied to impact in turbomachinery, they will be even more interesting!

Round 2

Reviewer 1 Report

Reviewer's  main comments have been addressed